



# Electron spin dynamics during microwave pulses studied by 94 GHz chirp and phase-modulated EPR experiments

Marvin Lenjer[1,2], Nino Wili[3], Fabian Hecker[1,4], and Marina Bennati[1,2]

[1]RG EPR Spectroscopy, Max Planck Institute for Multidisciplinary Sciences, Am Fassberg 11, 37077 Göttingen, Germany
[2]Institute of Physical Chemistry, Georg August University Göttingen, Tammanstrasse 6, 37077 Göttingen, Germany
[3]Interdisciplinary Nanoscience Center, Aarhus University, Gustav Wieds Vej 14, 8000 Aarhus C, Denmark
[4]Center for Hyperpolarization in Magnetic Resonance, Danish Technical University, Oerstedsplads 439, 2800 Kgs. Lyngby, Denmark

**Correspondence:** Marvin Lenjer (marvin.lenjer@mpinat.mpg.de)

**Abstract.** Electron spin dynamics during microwave irradiation are of increasing interest in electron paramagnetic resonance (EPR) spectroscopy, as locking electron spins into a *dressed* state finds applications in EPR and dynamic nuclear polarization (DNP) experiments. Here, we show that these dynamics can be probed by modern pulse EPR experiments that use arbitrary waveform generators to produce shaped microwave pulses. We employ phase-modulated pulses to measure Rabi nutations,
echoes, and echo decays during spin locking of a BDPA radical at 94 GHz EPR frequency. Depending on the initial state of magnetization, different types of echos are observed. We analyze these distinct coherence transfer pathways and measure the decoherence time $T_{2\rho}$, which is a factor $3-4$ longer than $T_{\mathrm{m}}$. Furthermore, we use chirped Fourier transform EPR to detect the evolution of magnetization profiles. Our experimental results are well reproduced using a simple density matrix model that accounts for $T_{2\rho}$ relaxation in the spin lock (tilted) frame. The results provide a starting point for optimizing EPR experiments
based on hole burning, such as electron-nuclear double resonance or ELDOR-detected NMR.

## 1 Introduction

Long microwave (MW) pulses with a duration of several microseconds are building blocks for pulse sequences in electron paramagnetic resonance (EPR) spectroscopy. They serve as preparation pulses to drive forbidden EPR transitions in hyperfine spectroscopy experiments like ELDOR-detected nuclear magnetic resonance (EDNMR) (Schosseler et al., 1994) and can also
lock spins into a *dressed* state, characterized by a distinct reorganization of their energy states (Cohen-Tannoudji and Reynaud, 1977; Jeschke, 1999). This enables electron-nuclear polarization transfer via cross-polarization (CP) under modified Hartmann-Hahn matching conditions as NOVEL or eNCP (Henstra et al., 1988; Weis et al., 2000; Weis and Griffin, 2006; Pomplun et al., 2008; Rizzato et al., 2013; Bejenke et al., 2020). Spin locking (SL) can also be harnessed to decouple spins from their surroundings, resulting in longer decoherence times of electron spins in EPR and optically detected magnetic resonance
due to decoupling of nuclear interactions (Jeschke and Schweiger, 1997; Rizzato et al., 2023). SL experiments were recently proposed for electron-electron distance determination, where the upper limit of detectable distances is determined by the





electron decoherence time (Wili et al., 2020). The altered energy landscape of electron spins locked by a MW field and under relaxation results in distinct spin dynamics, which are not yet fully explored.

One important aspect is the effect of decoherence during spin locking (i.e., $T_{2\rho}$ relaxation), which has recently been discussed
in the context of EPR spectroscopy by Wili et al. (2020). Some examples of treating this process exist in the literature. These approaches rely on a phenomenological description of decoherence by rate equations, but differ in the chosen interaction frame, which in turn determines the spin states that are affected by relaxation. For example, in a theoretical treatment of CP-ENDOR (Bejenke et al., 2020) we approximated the final state after a long SL pulse by propagating the spin density matrix in the dressed state yet without relaxation terms, followed by a heuristic implementation of $T_{2\rho}$ that eliminates off-diagonal matrix
elements. Hovav et al. (2010) described relaxation during MW irradiation in dynamic-nuclear polarization experiments using density matrix propagation in the eigenframe of the equilibrium Hamiltonian without MW irradiation. Similarly, Hajduk et al. (1993) and Cox et al. (2017) proposed to model the effect of NMR shaped pulses or EDNMR preparation pulses with the Bloch equations in the rotating frame, including $T_{\mathrm{m}}$ relaxation. In contrast, Desvaux et al. (1994), De Luca et al. (1999) and Michaeli et al. (2004) expressed relaxation rates during irradiation in the eigenframe of the complete Hamiltonian including
an irradiation term (i.e., the tilted frame) when analyzing the effect of dipolar interaction, molecular motion, and chemical exchange to the overall coherence decay rate.

This paper presents our efforts to gain insight in the electron spin dynamics of long MW pulses ($T_{2\rho} \lesssim t_{\mathrm{pulse}} \lesssim T_{1\rho}$). We focused on rationalizing decoherence effects and, ultimately, the trajectories of spins during MW irradiation to determine correct excitation profiles in EPR hole-burning experiments. Simultaneously, the experiments showcase the applicability of
shaped pulses at our commercial W-band spectrometer. In literature, most shaped pulse experiments were performed at MW frequencies lower or equal to Q-band (34 GHz) (Endeward et al., 2023). The use of shaped MW pulses at high frequencies was so far constrained to inversion pulses (Bahrenberg et al., 2017; Kuzhelev et al., 2018) or highly specialized experimental set-ups (Kaminker et al., 2017; Subramanya et al., 2022; Quan et al., 2023).

Experimentally, we followed two main routes. On the one hand, we performed pulse experiments during SL by coherently
manipulating the locked spins using periodic phase modulations (PM) of the locking field. (Grzesiek and Bax, 1995; Jeschke, 1999; Chen and Tycko, 2020; Wili et al., 2020). We report 94 GHz SL Rabi nutations, echos, and echo decays measured with pulse sequences established by Wili et al. (2020) that were analyzed with electron spin dynamic simulations. On the other hand, we applied chirp echo FT EPR spectroscopy (CHEESY) (Wili and Jeschke, 2018) to measure the excitation profiles of MW pulses in the regime between Rabi nutation behavior and steady-state conditions. Subsequently, we set up density matrix
simulations for an isolated electron spin under MW irradiation. Using the $T_{2\rho}$ values obtained from the PM experiments we were able to simulate the CHEESY excitation profiles. Comparison with simulations using the well-established *Spinach* library (Hogben et al., 2011) showed consistent results when using the suited interaction frame. Together with calculations based on the Bloch equations, we demonstrate that, for MW irradiation periods in the order of $T_{2\rho}$, calculating in the correct frame of reference is crucial for describing the experimental data.



## 2 Theorical description

In this section we provide the spin Hamiltonians used for the simulations of PM experiments and chirp echo profiles in their respective interaction frame. As will become evident from the results, a theoretical description based on the simple model of an isolated $S = 1/2$ electron spin under MW irradiation, including a MW frequency offset, is sufficient. We label laboratory frame operators without a prime, rotating frame operators with a single prime ($'$), and tilted frame (eigenframe during microwave irradiation) operators with a double prime ($''$). Operators in the nutating frame (interaction frame for PM pulses during spin lock) are denoted with an asterisk ($*$). Furthermore, we indicate how the time evolution of the density matrix in the different frames can be computed under the effect of decoherence. All equations are expressed in angular frequency units ($\omega$), while experimental results are reported in frequency units ($\nu = \omega/2\pi$) as recorded.

### 2.1 Spin Hamiltonian

The laboratory frame Hamiltonian $\hat{\mathcal{H}}$ for an isolated electron spin under irradiation with linearly polarized MW in $x$-direction contains the electron Zeeman and the microwave irradiation terms:

$$\hat{\mathcal{H}}(t) = \omega_0 \hat{S}_z + 2\omega_1 \cos(\omega_{\mathrm{MW}} t + \phi_{\mathrm{MW}}) \hat{S}_x \tag{1}$$

$\omega_0$ is the Larmor frequency, $\omega_{\mathrm{MW}}$ and $\phi_{\mathrm{MW}}$ are the MW frequency and phase, respectively, and $\omega_1$ is the Rabi frequency proportional to the MW $B_1$ field. To eliminate the time dependence, $\hat{\mathcal{H}}(t)$ is transformed into a rotating frame with $\hat{U}_1(t) = \exp(-i\omega_{\mathrm{MW}} t \hat{S}_z)$ (Abragam, 1961; Eckardt, 2017). Subsequent application of the rotating wave approximation (RWA) yields the time-independent rotating frame Hamiltonian:

$$\hat{\mathcal{H}}' = \hat{U}_1^{\dagger}(t) \hat{\mathcal{H}}(t) \hat{U}_1(t) - i\hat{U}_1^{\dagger}(t) \frac{\mathrm{d}\hat{U}_1(t)}{\mathrm{d}t} \tag{2}$$

$$\overset{\mathrm{RWA}}{\approx} \Omega_S \hat{S}_{z'} + \omega_1 \cos(\phi_{\mathrm{MW}}) \hat{S}_{x'} + \omega_1 \sin(\phi_{\mathrm{MW}}) \hat{S}_{y'} \tag{3}$$

where $\Omega_S = \omega_0 - \omega_{\mathrm{MW}}$ is the frequency offset of the MW to the electron Larmor frequency. Unless stated otherwise, MW irradiation is assumed to be applied along $y'$ (i.e., $\phi_{\mathrm{MW}} = 90°$) consistent with our experimental SL field.

#### 2.1.1 PM pulses and nutating frame

In the rotating frame, the Hamiltonian $\hat{\mathcal{H}}'_{\mathrm{PM}}$ during a PM pulse is defined as in Eq. 4, including a periodic variation of the phase $\phi_{\mathrm{MW}}(t)$ (Eq. 5) that is matched with $\omega_1$ (Wili et al., 2020).

$$\hat{\mathcal{H}}'_{\mathrm{PM}} = \Omega_S \hat{S}_{z'} + \omega_1 \cos(\phi_{\mathrm{MW}}(t)) \hat{S}_{x'} + \omega_1 \sin(\phi_{\mathrm{MW}}(t)) \hat{S}_{y'} \tag{4}$$

$$\phi_{\mathrm{MW}}(t) = \phi_0 + a_{\mathrm{PM}} \cos(\omega_{\mathrm{PM}} t + \phi_{\mathrm{PM}}) \tag{5}$$

$\phi_0$ is the spin lock phase, $a_{\mathrm{PM}}$ the modulation depth of the phase modulation and $\omega_{\mathrm{PM}}$ its frequency. $\phi_{\mathrm{PM}}$ is the phase of what will later be defined as the PM pulse. Assuming a small modulation amplitude $a_{\mathrm{PM}}$ and strong driving conditions (i.e.,





$\omega_1 \gg \Omega_S$), this expression can be transformed into the nutating frame, yielding Eq. 6 after application of the rotating wave approximation. (Grzesiek and Bax, 1995; Wili et al., 2020).

$$\hat{\mathcal{H}}_{\mathrm{PM}}^* \overset{\mathrm{RWA}}{\approx} \Omega_{\mathrm{d}} \hat{S}_{y^*} + \frac{\omega_1 a_{\mathrm{PM}}}{2} \left( -\cos(\phi_{\mathrm{PM}}) \hat{S}_{x^*} + \sin(\phi_{\mathrm{PM}}) \hat{S}_{z^*} \right) \tag{6}$$

Here, $\Omega_{\mathrm{d}} = \omega_1 - \omega_{\mathrm{PM}}$ is a nutating frame offset that is unavoidable due to MW inhomogeneity. In contrast to the experiments performed by Wili et al. (2020), our experiments do not fulfill the above-mentioned strong driving condition. Therefore, we used Eq. 4 for numerical simulations. Nevertheless, Eq. 6 provides an effective way of understanding PM experiments, demonstrating the analogy to a rotating frame experiment (see Eq. 3) with pulses along $\boldsymbol{x}^*$ or $\boldsymbol{z}^*$ and with a nutation frequency of

$\frac{\omega_1 a_{\mathrm{PM}}}{2}$. The validity of Eq. 6 under our experimental conditions is discussed in the results (see Sect. 4.1.1).

### 2.1.2 Tilted frame

During constant MW irradiation, a spin ensemble in the rotating frame is subject to an effective field $\boldsymbol{\omega}_{\mathrm{eff}} = \Omega_S \boldsymbol{z}' + \omega_1 \boldsymbol{y}'$ described by the vector sum of the spins offset and the microwave field. Therefore, $\hat{\mathcal{H}}'$ is not diagonal. Diagonalization of $\hat{\mathcal{H}}'$ leads to a rearrangement of the energy states due to the altered quantization axis (Rizzato et al., 2013) and is achieved by

a rotation of $\hat{\mathcal{H}}'$ with $\hat{U}_2$ around $\boldsymbol{x}'$ into a so called tilted or spin locked frame (Eq. 7). Note that, in this case, we apply the rotations on the operator and not the coordinate frame so that $\hat{\mathcal{H}}''(t)$ is obtained by Eq. 8 (Ernst et al., 1998).

$$\hat{U}_2 = \exp\left( -i\theta \hat{S}_{x'} \right) \tag{7}$$

$$\hat{\mathcal{H}}''(t) = \hat{U}_2 \hat{\mathcal{H}}'(t) \hat{U}_2^\dagger \tag{8}$$

In Eq. 7, $\theta = \mathrm{Tan}^{-1}(\omega_1/\Omega_S)$ is the tilt angle of the $\boldsymbol{z}''$-axis in the Hamiltonian eigenframe with respect to $\boldsymbol{z}'$, defined such

that $\theta \in [0°, 180°]$ for $\omega_1 > 0$. $\mathrm{Tan}^{-1}()$ is the four-quadrant inverse tangent (e.g., implemented in *MATLAB* as the *atan2*() command) that returns the correct quadrant of $\theta$ by using $\omega_1$ and $\Omega_S$ as separate input values (Bejenke et al., 2020). The transformation yields the tilted frame Hamiltonian $\hat{\mathcal{H}}''$ where $\boldsymbol{z}''$ is aligned with $\boldsymbol{\omega}_{\mathrm{eff}}$ (Eq. 9).

$$\hat{\mathcal{H}}'' = \omega_{\mathrm{eff}} \hat{S}_{z''} \tag{9}$$

This transformation is necessary to implement spin relaxation as discussed in the following.

### 2.2 Density operator and time evolution

The spin system evolution can be calculated using the density operator formalism. Here, the Hamiltonian ($\hat{\mathcal{H}}$, $\hat{\mathcal{H}}'$, $\hat{\mathcal{H}}''$ or $\hat{\mathcal{H}}^*$) acts on an ensemble of electron spins described by a time-dependent $2 \times 2$ density operator $\hat{\rho}(t)$ in the same frame of reference (i.e., $\hat{\rho}(t)$, $\hat{\rho}'(t)$, $\hat{\rho}''(t)$ or $\hat{\rho}^*(t)$). For the frame transformation of the density matrix, the previously defined transformation matrices are used with the same sense of rotation as for the corresponding Hamiltonian (Bejenke et al., 2020), e.g., $\hat{\rho}''(t) =$

$\hat{U}_2(t) \hat{\rho}'(t) \hat{U}_2^\dagger(t)$. The time evolution of any $\hat{\rho}$ is computed from the solution of the Liouville-von Neumann equation in the respective frame and under assumption of time-independent Hamilton operators, for example in the rotating frame:

$$\hat{\rho}'(t) = \hat{U}'(t) \hat{\rho}'(0) \hat{U}'^\dagger(t) \tag{10}$$





where $\hat{U}'(t) = \exp(-i\hat{\mathcal{H}}'t)$ is a time dependent propagator and $\hat{\rho}'(0)$ the initial density operator. To include relaxation into the time evolution of $\hat{\rho}'$, calculations are performed in Liouville space (Hovav et al., 2010). For this purpose, the Hamiltonian in Hilbert space is converted to the corresponding Liouvillian superoperator $\hat{\hat{L}}$ as shown in Eq. 11 for $\hat{\mathcal{H}}'$. The uppercase T denotes the transposed matrix and $\mathbb{1}$ is the $2 \times 2$ unit matrix (Kuprov, 2023).

$$\hat{\hat{L}}' = \mathbb{1} \otimes \hat{\mathcal{H}}' - \hat{\mathcal{H}}'^{\mathrm{T}} \otimes \mathbb{1} \tag{11}$$

The respective density operator in Hilbert space is converted into a four-element density vector operator $\hat{\boldsymbol{\rho}}(t)$. The ordering of the matrix elements into a vector is shown in Eq. 12 where $\alpha$ and $\beta$ are spin eigenstates (Ernst et al., 1987; Gyamfi, 2020).

$$\hat{\boldsymbol{\rho}}'(t) = \begin{pmatrix} \hat{\rho}'_{\alpha\alpha}(t) \\ \hat{\rho}'_{\alpha\beta}(t) \\ \hat{\rho}'_{\beta\alpha}(t) \\ \hat{\rho}'_{\beta\beta}(t) \end{pmatrix} \tag{12}$$

The evolution of the density vector operator is then evaluated using a propagation superoperator $\hat{\hat{U}}'(t)$ and the solution of the Liouville-von Neumann equation in the Liouville space:

$$\hat{\boldsymbol{\rho}}'(t) = \hat{\hat{U}}'(t)\hat{\boldsymbol{\rho}}'(0) = \exp\left(-i\hat{\hat{L}}'t\right)\hat{\boldsymbol{\rho}}'(0) \tag{13}$$

Starting from a Boltzmann populated equilibrium density operator $\hat{\rho}'(0) \propto \hat{S}_{z'}$, Eq. 13 allows numerical calculation of electron spin ensemble evolution under MW irradiation using matrix exponentials with standard mathematics software such as *MATLAB*. For better readability, we define $\hat{\rho}'(0)$ positive so that the sign of the quantum mechanical expectation values is the same as of corresponding macroscopic magnetization vectors (i.e., $\boldsymbol{M}(0) = (0,0,M_0)^{\mathrm{T}}$). Note that, due to the negative gyromagnetic ratio of the electron, the equilibrium density matrix in the high-temperature approximation is $\hat{\rho}(0) = \mathbb{1} - |R|\hat{S}_{z'}$ where $R$ is a thermodynamic population factor (Bejenke et al., 2020).

To describe relaxation *during* MW irradiation, relaxation terms are added in the tilted frame, thus, acting during propagation of the $\hat{\rho}''(t)$ density matrix. For this, a relaxation superoperator is defined using a phenomenological decoherence rate (Feintuch and Vega, 2017). In our simulations, we consider only transverse relaxation since the effect of longitudinal relaxation was found almost negligible on the timescale of our experiments. As relaxation occurs in the tilted frame, we name it $T_{2\rho}$ in agreement with literature (Desvaux et al., 1994; Michaeli et al., 2004; Wili et al., 2020), and the corresponding relaxation superoperator is $\hat{\hat{R}}_{2\rho}$.

$$\hat{\hat{R}}_{2\rho} = \begin{pmatrix} 0 & & & \\ & -\frac{1}{T_{2\rho}} & & \\ & & -\frac{1}{T_{2\rho}} & \\ & & & 0 \end{pmatrix} \tag{14}$$



Evolution of the spin system can then be calculated by solving the Liouville-von Neumann equation (Eq. 13) with a modified propagation superoperator $\hat{\hat{U}}''(t)$ that includes $\hat{\hat{R}}_{2\rho}$ (Feintuch and Vega, 2017).

$$\hat{\hat{U}}''(t) = \exp\left(\left(-i\hat{\hat{L}}'' + \hat{\hat{R}}_{2\rho}\right)t\right) \tag{15}$$

## 140   3   Materials and methods

### 3.1   EPR sample

A standard powder sample of $0.1\,\%$ $(m/m)$ 1,3-bisdiphenylene-2-phenylallyl (BDPA) in a polystyrene matrix was used for EPR experiments throughout this work. At W-band, the powder spectrum of this radical features a single Gaussian line with a full-width at half-maximum (FWHM) of approximately $8.8\,\mathrm{G}$ or $24\,\mathrm{MHz}$, due to non-resolved hyperfine couplings below

$10\,\mathrm{MHz}$ (see Fig. A7 (b)). Measurements were done at temperatures between $50$ and $100\,\mathrm{K}$.

### 3.2   Spectrometer configuration and characterization

All experiments were performed on a commercial Bruker E680 W-band EPR spectrometer equipped with an ENDOR resonator (Model EN600-1021H) and a liquid helium cryostat (Oxford Instruments). Pulse sequences were set up with the pulse programming language *PulseSPEL* implemented in Bruker's *Xepr* operating software. Shaped pulses were generated using

a Bruker *SpinJet* arbitrary waveform generator (AWG) with $1.6\,\mathrm{GS/s}$ sampling rate and $\pm 400\,\mathrm{MHz}$ bandwidth. In our set-up, W-band frequency is reached via a three-step, heterodyne mixing scheme. First, a constant frequency $\nu_\mathrm{x}$ in the range of $9.6 \pm 0.4\,\mathrm{GHz}$ is generated by a X-band source. Second, this carrier wave is mixed with the two AWG output channels $I(t)$ and $Q(t)$ for the real and imaginary part, respectively, using an IQ-mixer. Third, this shaped X-band pulse is mixed with the output of a phase-locked oscillator at $\nu_\mathrm{PLO} = 84.5\,\mathrm{GHz}$ to reach the pulse shape at W-band (see Sect. A1). As the AWG frequency is

centered at zero, the mixing is adjusted and filtering is applied to minimize the contribution of $\nu_\mathrm{LO} = \nu_\mathrm{x} + \nu_\mathrm{PLO}$ ('LO-leakage') and mirror frequencies. The pulses are amplified with a solid-state power amplifier (SSPA, Quinstar) with $2\,\mathrm{W}$ output power and transferred to the resonator through a circulator. Spin echoes are recorded after demodulation with $\nu_\mathrm{PLO}$ and subsequent amplification via a low-noise amplifier. Quadrature detection is done using $\nu_\mathrm{x}$ as a continuous reference.

To characterize our experimental conditions, we analyzed the phase stability of our set-up following Endeward et al. (2023).

The experiment (see Sect. A2) showed that the data suffers from phase-noise that can, however, be averaged out using multiple scans. We acquired the MW resonator profile in a range of $600\,\mathrm{MHz}$ using standard Rabi nutation experiments measured at different local oscillator frequencies $\nu_\mathrm{LO}$ (see Fig. A3 (a)). At $50\,\mathrm{K}$, we obtained a representative resonator bandwidth of approximately $140\,\mathrm{MHz}$ and a maximal Rabi frequency of $\nu_1 \approx 15\,\mathrm{MHz}$. Both values depend on the tuning conditions. Likewise, the amplification curve of the SSPA was measured with Rabi nutation experiments for the whole range of input

power levels. This experiment showed clear non-linearity of the MW amplification that was compensated during pulse shape generation. (see Fig. A3 (b)).





### 3.3 Experiments with phase-modulated pulses

If not denoted otherwise, all phase-modulated (PM) experiments during SL were initiated by a rectangular $\pi/2$-pulse immediately followed by a $90°$ phase shifted SL pulse. During this SL, the PM pulses were applied (see Fig. A4). The locked spins

were read out by a rectangular $\pi$-pulse in the rotating frame, after a delay $\tau$, which led to a spin echo at $2\tau$ after the SL pulse. Integration of this echo yielded the detected signal.

For each data point of a parameter sweep (e.g. $t_{\mathrm{PM}}$ or $\tau_2$, vide infra) during a PM experiment, a pulse shape file (I/Q values), including the spin lock as well as all PM pulses, was generated with a home-written *MATLAB* script, indexed and uploaded to the AWG. The complete experiment, including rectangular detection pulses and delays, was programmed in *PulseSPEL*.

Analogously, PM pulse phase cycling (PC) was achieved by generating individual pulse shapes for each combination of PM phases. Due to the memory limitations of the *SpinJet* AWG, the pulse shape files were generated with $10\,\mathrm{ns}$ time resolution of individual I/Q values, which was well below the AWG sampling rate but sufficient to encode the PM also for the maximal used modulation frequency with only minimal distortions of the pulse shape (see Fig. A4).

PM pulses were optimized with the experimental procedure reported by Wili et al. (2020). The modulation frequency $\omega_{\mathrm{PM}}$

was calibrated by monitoring the effect of a long PM pulse with low modulation depth $a_{\mathrm{PM}} = 0.04$ on the intensity of the final spin echo as a function of $\omega_{\mathrm{PM}}$. The PM frequency with the strongest effect was chosen for the following experiments as this corresponds to the matching condition $\omega_{\mathrm{PM}} = \omega_1$ (see Sect. A5). For the SL echoes, $a_{\mathrm{PM}} = 0.4$ was used and the PM pulse lengths were determined from the SL Rabi nutations.

### 3.4 Generation of chirp pulses

The chirp echo EPR experiments were set up based on the work by Doll and Jeschke (2014) and Wili and Jeschke (2018). Additional information can be found in the work by Endeward et al. (2023) and Spindler et al. (2016). Chirp pulses with a frequency range of 200 MHz and a length of $1\,\mu\mathrm{s}$ and $0.5\,\mu\mathrm{s}$ were generated with a home-written *MATLAB* routine. The pulse shapes featured a non-linear frequency sweep as a compensation for the previously measured resonator profile. This ensured approximately constant critical adiabaticity $Q_{\mathrm{crit}}$ during the frequency sweep (Baum et al., 1985; Doll and Jeschke,

2014) (see Sect. A6 for details). Additionally, the pulses were treated with a wide-band, uniform rate, smooth truncation (WURST) amplitude modulation, to reduce Fourier transform (FT) artifacts caused by the pulse edges (Kupce and Freeman, 1995). Compensation of the non-linearity of the SSPA yielded the final pulse shape. These pulse shape modifications are also implemented in the *pulse* function of *MATLAB*'s *EasySpin* toolbox (Stoll and Schweiger, 2006). Experimental inversion profiles of representative chirp pulses demonstrated that these pulses could achieve broadband excitation and inversion (Fig. A6).

We are aware that there are multiple possibilities of further optimizing the chirp pulses (Endeward et al., 2023). Most importantly, we measured the resonator profile only once and at a different temperature than used for most of the experiments shown in this work. Compensation based on this profile will lead to imperfections as the resonator profile is temperature and tuning dependent. Furthermore, there are other, more advanced compensation schemes that take into account the complete impulse response function of the spectrometer and not just its amplitude response (Doll et al., 2013; Endeward et al., 2023).



However, Doll and Jeschke (2017) showed that the effect of the latter is usually small. As this work is constrained to narrow-lined EPR samples where possible distortions caused by the chirp pulses are limited, we refrained from any further correction steps. Nevertheless, we expect that improved corrections might become necessary for samples with larger spectral width.

### 3.5 CHEESY experiments

Chirp echo EPR spectroscopy (CHEESY) experiments were set up following the Böhlen-Bodenhausen scheme. Namely, $1\,\mu s$
and $0.5\,\mu s$ chirp pulses were used as $\pi/2$- and $\pi$-pulse, respectively, to achieve simultaneous refocusing of spins with different resonance frequencies (Böhlen et al., 1989). An 8-step phase cycle of the chirp pulses ensured that only the desired signal was detected (see Sect. B4). BDPA spectra were measured at a frequency offset $\nu_{\text{off}} \approx -30\,\text{MHz}$ to avoid zero-frequency artifacts from LO-leakage and baseline imperfections.

The Rabi frequency of the chirp pulses was optimized by chirp nutations. For this purpose, the intensity and shape of
the Fourier-transformed chirp echo were monitored as a function of $\nu_1$ (Doll and Jeschke, 2014). The $\nu_1$ value where the integral of the FT spectrum was maximal was chosen as the optimal setting. To measure the spectral intensity across the whole $200\,\text{MHz}$ excitation bandwidth of the chirp pulses, a frozen sample of 4-hydroxy-2,2,6,6-tetramethylpiperidinyloxyl (TEMPOL) in $H_2O$/glycerol was used because it yielded a broad and featured FT spectrum. First, the Rabi frequency of the $\pi/2$-pulse was optimized using a non-optimized $\pi$ pulse. The experiment showed an optimum at $\nu_1 = 7.4\,\text{MHz}$ which,
integrated over the resonator profile, corresponded to $Q_{\text{crit}} \approx 0.6$. This value was close to the theoretical optimum of $Q_{\text{crit}} = 2\ln(2)/\pi \approx 0.44$ predicted by Jeschke et al. (2015) for a $\pi/2$ chirp pulse (see Fig. A7 (a)). Second, the optimized $\nu_1$ was set for the $\pi/2$-pulse, and the power of the $\pi$ pulse varied in the same way. The Rabi frequency with maximal signal was $\nu_1 \approx 5.6\,\text{MHz}$. Considering that this pulse was shorter than the $\pi/2$-pulse, its low optimal MW intensity was surprising. As discussed by Jeschke et al. (2015), this effect can be attributed to a combination of a dynamic $Q_{\text{crit}}$-dependent phase shift and
$\nu_1$ inhomogeneity. This observation suggested that the inversion efficiency of the pulse was low, presumably causing significant reduction of echo signal.

Despite this non-ideal conditions, we could achieve good agreement between the chirp echo FT spectrum of BDPA and the corresponding echo-detected (ED-) EPR experiment (Fig. A7 (b)). This demonstrated that we can detect narrow-lined EPR spectra using FT spectroscopy without creating distortions or artifacts. Additionally, we measured the offset dependence of this
performance, which showed that the FT intensity decreases with larger frequency offsets, approximately following the shape of the resonator profile (Fig. A7 (c,d)).

CHEESY experiments to determine pulse inversion profiles of rectangular pulses consisted of the analyzed pulse, followed by the optimized chirp echo. To generate a hole profile narrower than the EPR spectrum of BDPA, a low-power MW pulse ($\nu_1 \approx 0.7\,\text{MHz}$) was used. The CHEESY spectra were obtained by subtracting the background echo without an initial pulse
from the detected signal. The spectra were normalized to the background, and unity was added to afford the $z'$-magnetization profiles, as demonstrated by Wili and Jeschke (2018). In contrast to the detection of pulse profiles with ELDOR (Hovav et al., 2015; Zhao et al., 2023), this procedure yielded the complete hole shape in a single experiment without any parameter sweep.



Similar experiments for the analysis of inversion profiles via FT were reported by Tait and Stoll (2017), yet without the use of chirped pulses.

### 235  3.6  R-CHEESY

We performed refocused (R)-CHEESY experiments with a modified pulse sequence consisting of two chirped $\pi$-pulses of equal length, which refocused coherence generated by the preceding analyzed pulse. In analogy to the Böhlen-Bodenhauser scheme for the standard chirp echo, the second refocusing chirp pulse was required to refocus spins with different offsets at the same time (Böhlen et al., 1989; Jeschke et al., 2015). Both pulses of this sequence were set with an optimized MW power,

as described in the previous section. To filter all undesired echoes, a 32-step phase cycle was used (Bodenhausen et al., 1977; Cano et al., 2002, see Sect. B5). Without a background signal for reference, all spectra of one data set were normalized to the global maximum.

As the FT provided a complex signal, the coherence's $x'$ and $y'$ components could be separated in the real and imaginary signal parts. However, in contrast to the $z'$-profiles where phasing could be done with respect to the Gaussian line of the

unperturbed BDPA spectrum, the $x'y'$-profiles only contained the coherence generated by the analyzed pulse. This prohibited an absolute phase determination, as there was no reference phase for the data processing. Therefore, the phase was adjusted for the individual FT spectra to maximize the absorptive (i.e., axially symmetric to zero) signal in the real channel. In this way, we achieved agreement with simulations (vide infra).

### 3.7  Fourier transformation of chirp echoes

Chirp echoes obtained from CHEESY spectroscopy were offset-corrected with a zeroth order baseline correction. A symmetric FT window around the echo signal was chosen, and the experimental signal was apodized with a cosine window function. The result was zero-filled with the number of zeros equal to the number of data points and shifted using the *fftshift* command of *MATLAB*. This time-domain signal was Fourier transformed using the fast Fourier transformation algorithm implemented in *MATLAB*. Subsequently, a constant phase correction with $\phi_0$ was applied to afford the phased, complex FT spectrum. If not

denoted otherwise, $\phi_0$ was selected to maximize the absorptive contribution in the signal's real part.

For the CHEESY $z'$-profiles, this process could be automated using a home-written *MATLAB* routine. In the case of the R-CHEESY, the FT window had to be selected manually for each data set because the automatized selection of the time-domain signal maximum was unstable. If not adjusted manually, this led to linear phase drifts in the resulting FT spectra.

### 3.8  Spin density matrix simulations

The home-written simulations reported in this work were performed based on the density operator formalism in Liouville space. The initial density operator $\hat{\rho}'(0)$ for a $S = 1/2$ electron spin system was defined according to Sect. 2.2. The Hilbert space Hamiltonian for each piece-wise constant element of the simulated sequence was set-up in the selected frame of reference. If necessary, $\hat{\rho}'(0)$ was transferred into the same reference frame. Both the initial density matrix and the Hamiltonians were then





transformed into their respective Liouville space representations and the latter used to calculate the corresponding propagators
(Eq. 11−13). The density matrix was propagated by consecutive application of the propagation superoperators (Eq. 13) and
the final output was obtained as the expectation value of the Cartesian operator of interest. Simulation parameters were set in
agreement with their experimental counterparts.

SL Rabi nutations (Sect. 4.1.1) and SL echoes (Sect. 4.1.2) were calculated in the rotating frame using $\hat{\rho}'(0) = \hat{S}_{y'}$ or $\hat{S}_{z'}$
for SL and without-preparation (WOP) SL experiments, respectively, and assuming $\Omega_S = 0$. While the delays between the
PM pulses were simulated in a single step using $\hat{\mathcal{H}}'$ as defined in Eq. 3, the PM pulses were incremented with a step size
$\Delta t = 1\,\text{ns}$ because $\hat{\mathcal{H}}'_{\text{PM}}$ is time dependent. In this way, a constant Hamiltonian could be assumed within these increments, as
demonstrated by Laucht et al. (2016). Relaxation was neglected during these calculations. For each value of $t_{\text{PM}}$ or $\tau_2$, the
result was obtained as the expectation value $\langle \hat{S}_{y'} \rangle$ (see Fig. 1 (c)). In case of the SL echo experiments, MW inhomogeneity was
included in the SL echo simulations by repeating the calculation for different $\nu_1$ values. The sum of these individual traces
weighted with an experimental $\nu_1$ profile yielded the depicted simulation results (details in Sect. A5). To calculate the effect of
PC, the simulations were repeated with different phases $\phi_{\text{PM2}}$ and the results combined in the same way as in the experiment.

The CHEESY profiles were simulated starting from $\hat{\rho}'(0) = \hat{S}_{z'}$. A linear array of spin offsets $\Omega_S/2\pi$ was generated cor-
responding to the experimental $\nu - \nu_{\text{off}}$ axis. For each value of $\Omega_S$, the initial density matrix $\hat{\rho}'(0)$ was transformed into the
tilted eigenframe of the MW Hamiltonian $\hat{\mathcal{H}}'$ (Eq. 3,7 and 8). The final state after a pulse with length $t_{\text{p}}$ was calculated using a
single propagator $\hat{U}''$ calculated from $\hat{\mathcal{H}}''$ (Eq. 9) and including $T_{2\rho}$ relaxation (see Eq. 14 and 15). Again, MW inhomogeneity
was included as the weighted sum of the results for different $\nu_1$ values using the experimental $\nu_1$ profile shown in Sect. A5.
After back-transformation of $\hat{\boldsymbol{\rho}}'(0)$ into the rotating frame, the final result was obtained as the expectation values $\langle \hat{S}_{z'} \rangle$ for the
CHEESY $z'$-profiles and as $\langle \hat{S}_{x'} \rangle$ and $\langle \hat{S}_{y'} \rangle$ for the $x'y'$-profiles (see Fig. 4 (e)).

### 3.9 Spinach simulations

Simulations using the Spinach library (Hogben et al., 2011) were done based on an adapted version of the *holeburn.m* function.
The CHEESY spectrum was calculated for an anisotropically broadened powder EPR line (see Sect. A13) in three steps. First,
the initial equilibrium spin system was propagated under a soft MW SL pulse including transversal relaxation, in the tilted
frame. Second, an ideal, non-selective $\pi/2$-rotation was applied that transformed all residual $z'$-components into coherences.
Third, simulation of the powder averaged free induction decay (FID) and FT yielded the CHEESY spectrum. In order to filter
the FID of the SL pulse, a 2-step PC of the SL was applied in the simulations (i.e., $\phi_{\text{SL}} = [0°, 180°]$ and $\phi_{\text{det}} = [+, +]$). Due
to the increased computational cost of the calculations, $\omega_1$ inhomogeneity was not included. The results were compared to
calculations using the unmodified *holeburn.m* function and rotating frame relaxation implemented as *t1_t2.m* (see Sect. 4.2.1
and A13).





## 4 Results

### 4.1 Spin locked EPR experiments

#### 4.1.1 SL Rabi nutations

Figure 1 (a) shows the pulse sequence for SL Rabi nutation experiments, which can be used to determine the optimal PM pulse lengths for further SL EPR experiments. The sequence consists of an initial rectangular $\pi/2$ pulse $(x')$ that prepares electron coherence, which is subsequently locked by the SL pulse along $y'$. Preparation and SL pulse were performed with $\nu_1 = 15\,\mathrm{MHz}$, which was considerably less than the $100\,\mathrm{MHz}$ used by Wili et al. (2020) at 34 GHz and not sufficient to lock the entire BDPA spectrum with a width of $\approx 24\,\mathrm{MHz}$ in the transversal plane. The effects of this will be discussed later in this section.

During SL and after an initial SL delay $\tau_0$, a PM pulse was applied by periodic modulation of the SL phase for a variable length $t_{\mathrm{PM}}$ and with a modulation amplitude $a_{\mathrm{PM}}$. The overall SL length $\tau_0 + \tau_1$ was constant. The effect of the PM pulse was monitored by a bare state (rotating frame) echo produced by a $\pi$-pulse after a delay $\tau$ that refocused coherence in the rotating frame at $2\tau$ after the SL pulse. Figure 1 (b) shows the SL Rabi nutation traces for three different modulation amplitudes $a_{\mathrm{PM}}$. The SL Rabi traces contained a dominant slow oscillation with a superimposed weaker and faster modulation. FT of the time traces (Fig. 1 (d)) showed that the dominant, slower oscillation corresponded to a SL Rabi frequency of $\nu_1 a_{\mathrm{PM}}/2$. This agrees with the theoretical predictions from the nutating frame transformation (Eq. 6). The faster oscillation frequency appeared at $2\nu_1$ plus two sidebands (Fig. 1 (e)).

We simulated the experiments using density matrix calculations (see Fig. 1 (c) and Sect. 3.8), neglecting a bare state offset by setting $\Omega_S = 0$. Figure 1 (d) and (e) show almost quantitative agreement between the Fourier transformed experimental and simulated time traces. Interestingly, the fast oscillation also directly resulted from the simulation. We attribute this effect to a deviation from the RWA, leading to a minor, yet visible modulation with the counter-rotating wave that precesses with $2\nu_1$. The simulations also qualitatively reproduced the sidebands of $2\nu_1$, which might be caused by frequency mixing. Comparison with the results by Laucht et al. (2016) shows that they reported similar fast frequency oscillations in their simulation for tilted frame excitation by modulation of the frequency offset instead of phase modulations.

This good agreement without simulating any bare state offset was unexpected, as the strong driving condition $\omega_1 \gg \Omega_S$ is not met in our experiments. Simulations including a bare state offset showed that severe distortions occurred for a single, sharp offset $\Delta\Omega > \omega_1$ (see Fig. A9 (a)). However, simulating a Gaussian distribution of $\Omega_S$-values that was centered at $0\,\mathrm{MHz}$ yielded a nearly unperturbed FT spectrum (Fig. A9 (b)). This suggests that offset effects are averaged during the experiment, so that the final spectrum agrees with the predictions from the ideal $\omega_1 \gg \Omega_S$ case.

#### 4.1.2 SL echoes

In the next step, we examined the feasibility of observing spin echoes during SL. Using a PM $\pi/2$-pulse followed by a delay and a PM $\pi$-pulse, the analog of a Hahn echo can be created during SL (Wili et al., 2020). The refocused SL coherence is read

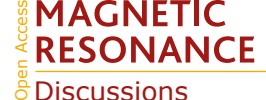

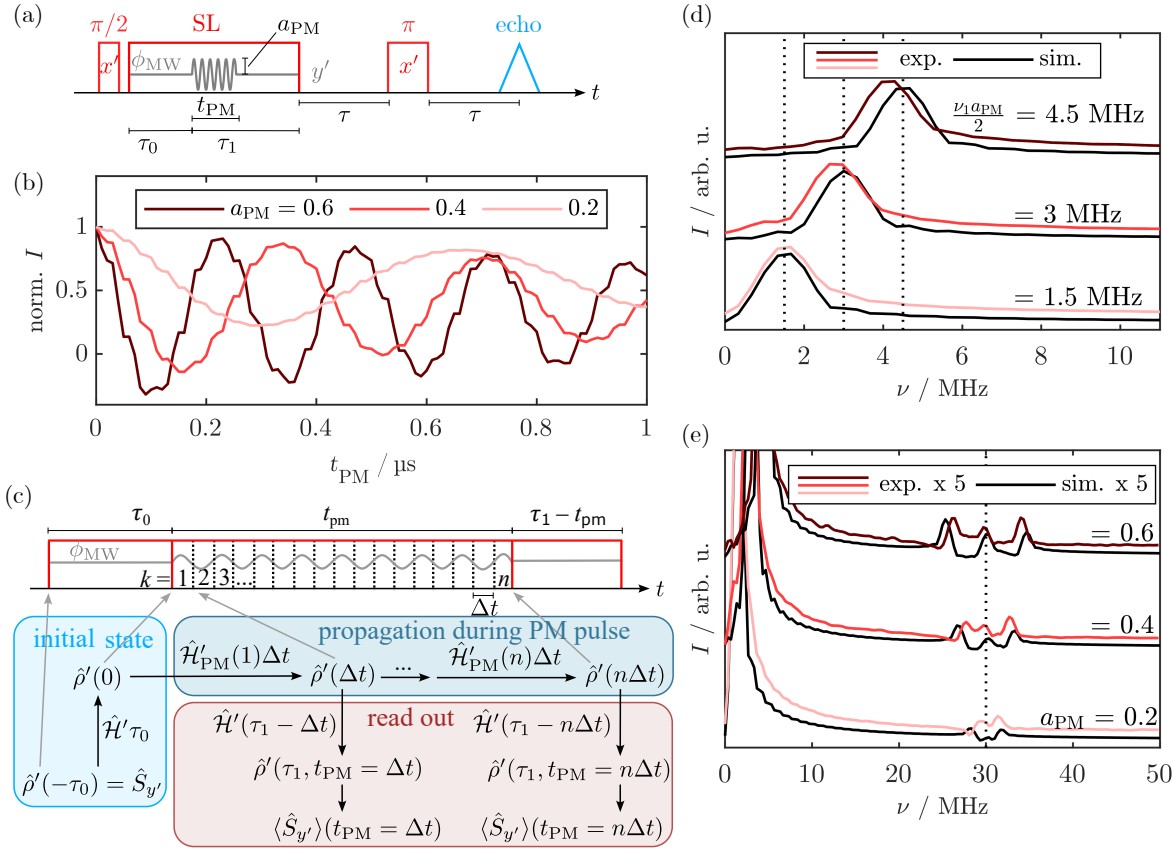

**Figure 1.** (a) Pulse sequence for the SL Rabi nutation. The echo was integrated as a function of the PM pulse length $t_{PM}$. (b) Experimental SL Rabi nutation traces for $\nu_1 = 15$ MHz with different modulation amplitudes $a_{PM}$. (c) Simulation approach for the SL Rabi nutation experiments by step-wise evolution of the rotating frame density operator during a phase-modulated pulse. (d,e) FT spectra of the experimental nutation traces (red) in the low-frequency region (d) and enlarged in the high-frequency region (e). Simulated nutations are shown in black, accurately reproducing the experimental peak positions. In (d), dotted lines mark the positions where a signal is expected for the respective SL Rabi frequency $\nu_1 \cdot a_{PM}/2$. The dotted line in (e) marks $\nu = 2\nu_1$. For experimental parameters, See Sect. B1.

out by a third PM $\pi/2$-pulse that realigns the magnetization with the SL field, which can be refocused with a bare state echo (see Fig. 2 (a)). We monitored the refocusing of SL coherence by variation of the delay $\tau_2$ between the refocusing (PM2) and read-out pulse (PM3). The SL pulse length was kept constant by simultaneous variation of $\tau_3$. For $\tau_2 = \tau_1 = 3$ μs, refocusing was observed as a strong echo (Fig. 2 (c)), which indicates that PM experiments are feasible under W-band experimental

330 conditions, i.e., weaker MW pulses and considerable offset contributions. In addition to this, weak signals (Fig. 2 (c), asterisks) whose position depended on the length of the delay $\tau_0$ were observed. According to Wili et al. (2020), these are caused by incomplete preparation of the coherent state $S_{y'}$, leaving spins with residual $S_{z'}$ components unaligned with the spin lock field.





In the same way as the $\pi/2$ preparation pulse, the PM pulse rotation angles can deviate from the desired $\pi/2$ and $\pi$ values, so that several signals can be produced by different effective rotation angles of the individual PM pulses. We used 2- and 4-step phase-cycling (PC) of the phase-modulated inversion pulse PM2 to eliminate the small crossing echoes and 2-step phase-cycling of the phase-modulated readout pulse PM3 to filter non-locked components. A 2-step PC of PM2 ($\phi_{\mathrm{PM2}} = [0°, 180°]$) with opposite detection phase ($\phi_{\mathrm{det}} = [+, -]$) selects coherence transfer pathways in which PM2 changes the coherence order by one, i.e., only when it acts as a $\pi/2$ pulse. In contrast, a 4-step PC of PM2 ($\phi_{\mathrm{PM2}} = [0°, 90°, 180°, 270°]$ with alternating detection phase ($\phi_{\mathrm{det}} = [+, -, +, -]$), selects a transfer pathway, where PM2 causes a change in coherence order by two, i.e., where it acts as a $\pi$-pulse. As expected, the 2-step PC of PM2 eliminated the main echo signal, whereas it was retained by the 4-step PC (see Fig. 2 (c)). Additionally, the small crossing echoes were affected by the different phase cycles, demonstrating that they arise from different coherence transfer pathways.

As incomplete preparation is supposed to be significant under W-band conditions, we analyzed the SL coherence pathways by omitting the $\pi/2$ preparation step. This type of experiments without preparation (WOP) was already implemented for CP-ENDOR by Bejenke (2020) but the spin dynamics were not reported. Figure 2 (b) depicts the modified pulse sequence, and (c) and (d) show the comparison of the SL echo with and without preparation obtained by variation of $\tau_2$ while keeping $\tau_0 = 2\,\mu s$ and $\tau_1 = 3\,\mu s$ fixed. Both experiments were performed using the same experimental parameters.

The WOP SL echo experiment produced signals at $1\,\mu s$ (ii), $2\,\mu s$ (iii) and $5\,\mu s$ (iv) (Fig. 2 (d)), which differs significantly from those observed in (c). Notably, the primary echo (i) is missing, while the observed echo positions coincide with the artifacts observed in (c). To rationalize this observation, we analyzed the refocusing times $\tau_2$ and the effect of PC. The strongest WOP SL echo (iii) is observed at $\tau_2 = \tau_0$, suggesting that it is created by a coherence that starts to evolve during $\tau_0$ and refocuses as a stimulated echo. This is corroborated by the PC, as the signal is retained by a 2-step PC of PM2 and vanishes upon 4-step PC. With the same assumption, signals (ii) and (iv) can be assigned to a refocused echo and Hahn echo, respectively. Coherence diagrams visualizing the idealized evolution pathways (i.e., $\Omega_S = 0$) responsible for each signal are shown in Fig. 2 (e), along with their respective refocusing condition. These results are consistent with the spins having a Boltzmann distribution in the rotating frame corresponding to $z'$-magnetization prior to the SL. Upon start of the SL pulse, i.e., under MW irradiation, this becomes a coherence in the tilted frame, which immediately evolves with $\nu_1$.

The peak positions (i)−(iv) as well as their line width and the effect of PC could be reproduced by density matrix simulations following the procedure described in Sect. 3.8 and Fig. 1 (c). The phase and intensity of the WOP signals agreed only partially. Repetition of the experiment under nominally same conditions showed that the phase of the individual signals was not reproducible. Detection of the out-of-phase component by setting $\phi_{\mathrm{PM3}} = \pi/2$ underlined that there is a phase instability of the WOP echoes (see Sect. A9). We suppose that this is the case because the turning angles of the PM pulses was not optimized for these specific coherence pathways.

Altogether, our results are consistent with the assumption that equilibrium $z'$-magnetization becomes a coherence during MW irradiation. Hence, these results show that, under our experimental conditions, conventional spin locking with a preparation step and long MW pulses acting on spins initially in thermal equilibrium are closely related. By application of PM pulses, the spin states can be transferred from SL coherence to SL populations and vice versa.

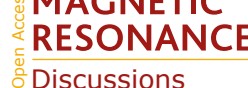

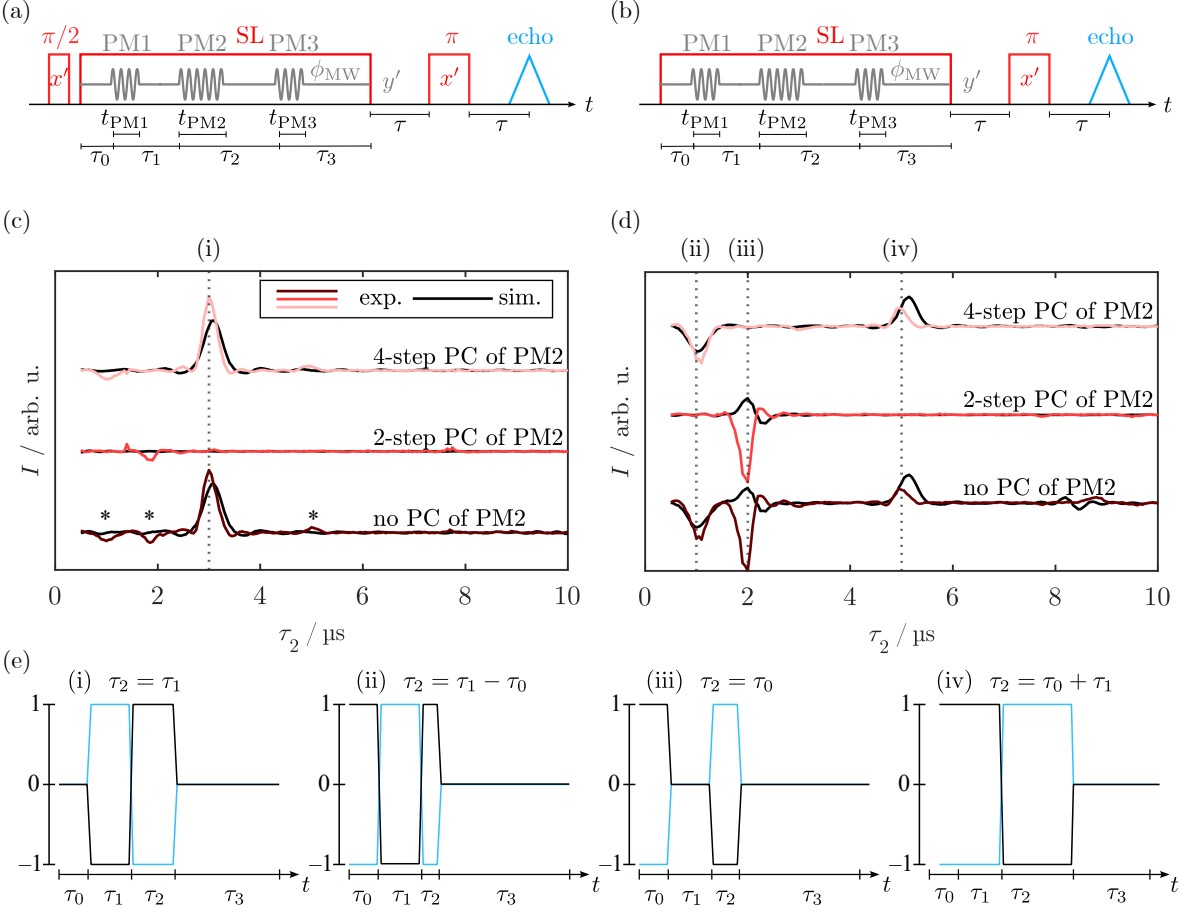

**Figure 2.** Pulse sequence for standard (a) and WOP (b) SL echo. Real component of the quadrature detected signal of the SL echo (c) and WOP SL echo (d) spectra measured with different phase cycles. The detection phase was set to maximize $y'$ magnetization in the real channel. Experimental traces in different shades of red, corresponding simulations in black. Signals (i) − (iv) were assigned in (e) and belong to evolution pathways that are refocused at the beginning of PM3. In (c), the asterisks denote the residual signals (ii), (iii) and (iv). The imaginary part of the spectra with signals refocused at the end of $\tau_3$ can be found in Fig. A10. For experimental parameters, see Sect. B2.

### 4.1.3 Relaxation during spin lock

To measure relaxation times during SL, the delays $\tau_1$ and $\tau_2$ of the SL echo sequence (see Fig. 2 (a)) were both increased
so that refocusing was achieved throughout the experiment while the evolution time in the transverse plane was increased. Simultaneously, $\tau_3$ was decreased to ensure a constant SL length. The comparison of the decays obtained with $\nu_1 = 15\,\text{MHz}$ and the bare state Hahn echo decay can be found in Fig. 3 (a). The experimental traces were fitted using both a mono- and a stretched exponential function. We calculated the residuals and statistical errors were determined by bootstrapping (for details, see Sect. A10). With this, we could compare the two decay models and evaluate the systematic error introduced when using the





simplified mono-exponential model. The decay of coherence was approximately three times slower in the SL state than in the rotating frame with mono-exponential decay times of $3.9 \pm 0.2\,\mu s$ and $1.36 \pm 0.02\,\mu s$, respectively (red and blue curve in Fig. 3 (a)). Fitting with a stretched exponential decay function led to $T_{2\rho} = 5.0 \pm 0.3\,\mu s$ and $T_{\mathrm{m}} = 1.46 \pm 0.01\,\mu s$. Analysis of the residuals (bottom part in Fig. 3 (a)) showed that there is a clear systematic deviation in the mono-exponential fit (red curve). Namely, the intensity is overestimated for $t_{\mathrm{p}} < 10\,\mu s$ and underestimated for $t_{\mathrm{p}} > 10\,\mu s$. This systematic error was reduced

when using the stretched exponential fit (dark red curve), which is consistent with literature concerned with bare-state echo decays, where stretched exponential functions are often used. (Mims et al., 1961; Jahn et al., 2022).

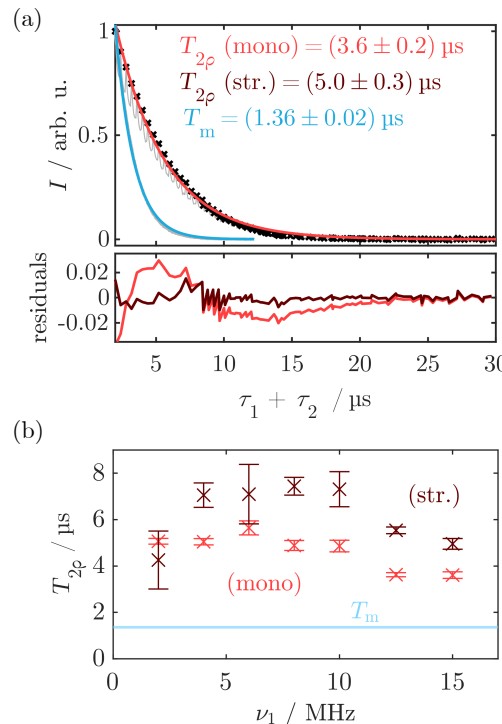

**Figure 3.** SL relaxation measurements obtained with the standard SL echo sequence as depicted in Fig. 2 by variation of $\tau_1 + \tau_2$. (a) SL relaxation trace measured for $\nu_1 = \nu_{\mathrm{PM}} = 15\,\mathrm{MHz}$ (red) compared to the $T_{\mathrm{m}}$ relaxation trace in the bare state (blue). Experimental time traces in gray and envelope function used for fitting the SL state decay as black dots. Mono-exponential fits for $T_{2\rho}$ and $T_{\mathrm{m}}$ in red and blue, respectively. Residual of the mono-exponential fit for $T_{2\rho}$ in the bottom part (red) compared to the residuals from the stretched-exponential fitting (dark red). (b) $T_{2\rho}$ data measured for different SL powers. The corresponding traces and fits can be found in Fig. A12 (a). Error bars correspond to the statistical error, while the systematic error might be larger. For experimental parameters, see Sect. B3.

Investigation of the power dependence of $T_{2\rho}$ (Fig. 3 (b) and A12) showed no clear trend and an overall increase of the transverse relaxation time by a factor three to five compared to the bare state was obtained from a mono exponential fit (red). The stretched exponential fitting yielded $T_{2\rho}$ values between $4\,\mu s$ and $8\,\mu s$. These results agree well with the low-field results

by Wili et al. (2020) that were performed at much higher mw fields $\nu_1 \approx 100\,\mathrm{MHz}$, reporting an increase of $T_{2\rho}$ relative to





$T_\mathrm{m}$ by a factor of $4.5$ for stretched exponential fitting. These observations show potential for designing new pulse sequences that mitigate short decoherence times. The stretching exponents $\xi$ for the stretched exponential fits were between $0.9$ and $1.7$ (see Sect. A10). Importantly, there is a strong correlation between the $T_{2\rho}$ value and $\xi$ (Pearson correlation within the bootstrap samples above $0.9$ for all traces). Therefore, the choice of the stretching coefficient affected the obtained decay time.

Notably, the SL echo decays were perturbed by an oscillation for small $\tau_1 + \tau_2$ (gray curve) which, to our knowledge, has not been reported in the literature. We attribute this to the edge effects of the PM pulses. A detailed analysis and quantitative explanation of these artifacts can be found in Sect. A11 but is of little consequence for the following results. We circumvented this effect by fitting the envelope function of the time trace (black dots in Fig. 3 (a)). This method is also used to analyze Hahn echo decay traces perturbed by electron spin echo envelope modulations (ESEEM) (Soetbeer et al., 2021). However, the

periodic oscillations in the residuals for $\tau_1 + \tau_2 < 10\,\mu\mathrm{s}$ are indicative of small artifacts caused by this envelope fitting.

### 4.2   CHEESY detection of pulse excitation profiles

#### 4.2.1   $z'$-profiles

We probed the residual $z'$-magnetization after a soft MW pulse of varied length $t_\mathrm{p}$ with a chirp echo (see Fig. 4 (a)). For each $t_\mathrm{p}$, FT of the chirp echo transient and subtraction of a background chirp echo without additional pulse yielded the inversion

profile as a hole burned in the initial $z'$-magnetization (Fig. 4 (b)). Figure 4 (c) shows the obtained hole shape as a function of $t_\mathrm{p}$ (red).

For short irradiation times, the profiles were sinc-shaped, as expected for a rectangular pulse. A distinct Rabi oscillation was detected, i.e., a periodic variation of the hole depth at the center frequency $\nu - \nu_\mathrm{off} = 0$ as a function of $t_\mathrm{p}$ (Fig. 4 (d), red). FT of the Rabi oscillation trace yielded a nutation frequency $\nu_1 = 0.77\,\mathrm{MHz}$. The magnitude of the oscillation decreased with

increasing pulse length. Comparison with the experimental $T_{2\rho}$ relaxation traces and simulations (see Fig. A14 (c)) showed that this decay was faster than the SL coherence decay. We attribute this fast decay to MW inhomogeneity, resulting in a broadened distribution of nutation frequencies. Simulations including both the experimentally determined $T_{2\rho}$ relaxation rate and MW inhomogeneity could quantitatively reproduce the Rabi trace (black in Fig. 4 (d), see Sect. A12). For long MW pulses with $t_\mathrm{p} \gtrsim 10\,\mu\mathrm{s}$, a steady state was reached where the excitation profile no longer depended on $t_\mathrm{p}$ and resembled a Lorentzian-

shaped saturation profile with $\mathrm{FWHM} \approx 1.6\,\mathrm{MHz}$. This saturation hole shape agrees with observations of the central hole in standard EDNMR experiments with rectangular high-turning angle pulses and a full-width half-maximum of $2\nu_1$ (Nalepa et al., 2014).

The simulations also reproduced the transition from a $t_\mathrm{p}$-dependent, sinc-shaped hole to a steady state Lorentzian as well as the excitation bandwidth. Moreover, the dampening of the Rabi oscillation curve was simulated quantitatively. Crucially, the

simulation results changed when the transformation in the tilted frame was omitted. In this case, the simulations for long $t_\mathrm{p}$ no longer matched the experiment, and an increased width of the simulated hole was observed (light blue in Fig. 4 (c)). We attribute this to the fact that, in this case, the relaxation term in the Liouville space propagator does not connect the correct states, i.e., the eigenstates of $\hat{\mathcal{H}}$, which leads to incorrect dampening during relaxation.

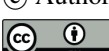



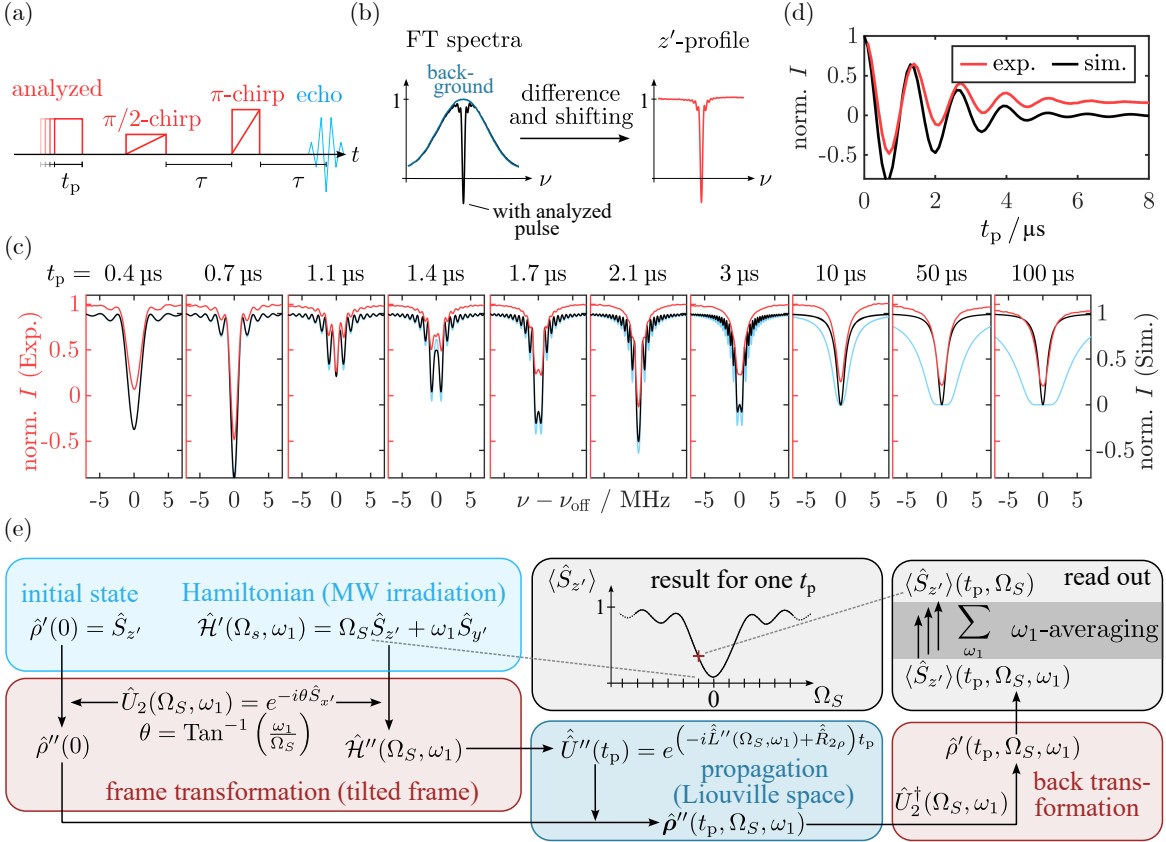

**Figure 4.** (a,b) CHEESY pulse sequence and schematic data processing. Inversion ($z'$-) profiles were obtained by subtracting a chirp echo background spectrum without the first pulse from the normalized hole profiles and vertically shifting the results by one. (Wili and Jeschke, 2018) (c) Experimental (red) and simulated (black) inversion profiles for a low power MW pulse ($\nu_1 = 0.77\,\mathrm{MHz}$). Simulations were performed using the simulation routine depicted in (e) with a variation of $t_\mathrm{p}$ and $\Omega_S$ (see Sect. 3.8). Results from an analogous simulation omitting the tilted frame transformation are shown in light blue. (d) Rabi nutation obtained from CHEESY inversion profiles at $\nu - \nu_\mathrm{off} = 0$ (red) and from tilted frame simulation (black). For experimental parameters, see Sect. B4.

The most apparent difference between experimental and simulated profiles in the tilted frame is the resolution in the frequency dimension, i.e., the degree to which the narrow sinc lobes for intermediate pulse lengths are resolved. While the oscillations are clearly visible in the simulations up to $t_\mathrm{p} = 3\,\mathrm{µs}$, they are hardly visible in the experimental spectra with $t_\mathrm{p} > 1.6\,\mathrm{µs}$. This is presumably caused by the intrinsic experimental FT resolution limit posed by the finite FT window during data processing.

We compared our results to simulations using the Spinach library (Hogben et al., 2011). As in the home-written simulation routine, transformation in the tilted frame during propagation of the SL pulse was necessary to reproduce the experimental





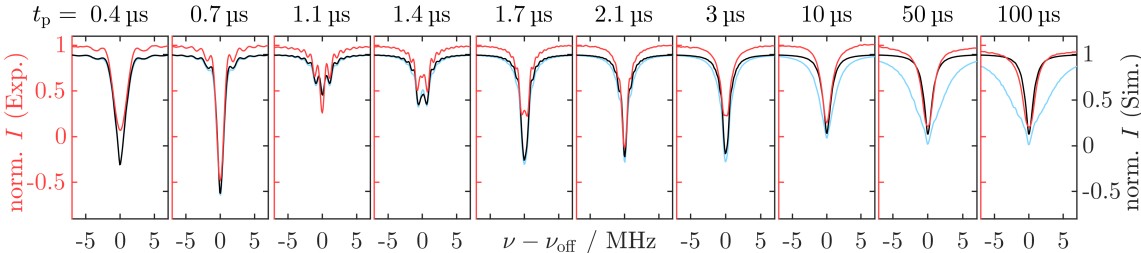

**Figure 5.** CHEESY z'-profiles simulated with the Spinach library (Hogben et al., 2011) using propagation in the tilted frame (black) and comparison to the experimental profiles (red). Analogous calculations in the rotating frame using the holeburn.m function in light blue. For experimental parameters, see Sect. B4.

results (compare black and light blue in Fig. 5). Compared to Fig. 4, the line width was reproduced more accurately because the FT was explicitly included in the simulation (see Sect. 3.9).

Comparison of simulations with different $T_{2\rho}$ values showed that using a shorter relaxation time $T_{2\rho} = 1.4\,\mu s$ identical to $T_m$ could not reproduce the data because the decay of the side lobes was too fast. Using a longer relaxation time $T_{2\rho} = 10\,\mu s$

did not affect the visual shape of the pulse profiles. In contrast, completely omitting $T_{2\rho}$ could not reproduce the experiment because no steady state was reached, even for $t_p = 100\,\mu s$ (see Fig. A15).

So far, our simulations neglected hyperfine coupling. This is a strong simplification, as nuclear coupling, in particular its pseudo-secular component, is known to introduce additional evolution pathways during irradiation (Jeschke, 1996). Using the Spinach implementation, nuclear coupling could be introduced straightforwardly. Simulation with up to four coupled protons

with $A < 10\,\text{MHz}$ led to no change in the CHEESY profiles when using an appropriately expanded relaxation superoperator in the tilted frame (see Fig. A16 (c)).

#### 4.2.2 $x'y'$-profiles

The standard CHEESY approach is limited to observing $z'$-magnetization and hole-burning profiles. To observe the coherence generated by a MW pulse, we used a modified chirp sequence where the chirp echo was replaced with two chirped $\pi$-pulses

that form, together with the analyzed pulse, a refocused echo sequence (R-CHEESY, see Fig. 6 (a)). This experiment is closely related to the ABSTRUSE sequence developed by Cano et al. (2002). FT of the hereby generated Chirp echo yielded a complex signal. Its phase was adjusted so that the absorptive signal was maximized in the real component.

The results of the R-CHEESY experiment are shown in Fig. 6 (c, red). Oscillating components were observed for short pulse lengths $t_p$ that changed to a steady state for longer irradiation in the same manner and time scale as observed in the CHEESY

$z'$-profiles. Monitoring the intensity of the real component at $\nu - \nu_{off} = 0$ showed a continuous Rabi oscillation (Fig. 6 (d)). FT of this trace yielded the Rabi frequency $\nu_1 = 0.63\,\text{MHz}$ (these experiments were performed with slightly different tuning of the microwave cavity and different power settings compared to the $z'$-profiles in the previous section). After long MW irradiation ($t_p \gg T_{2\rho}$), the steady state exhibited negligible signal in the real part, yet a clear dispersive signal was detected in



the imaginary component. This corresponded to no detectable magnetization perpendicular to the MW axis but to a distinct

magnetization component parallel to $\boldsymbol{\nu}_{\mathrm{eff}}$. To rationalize this, we simulated the $x'y'$-profiles using the procedure described in Fig. 4 (e) and Sect. 3.8. Again, there was a good agreement between experimental and simulated data for both the initial Rabi oscillation period and the steady state (Fig. 6 (c,d), black). As in the experiment, no contribution remained in the real part of the steady state R-CHEESY signal, while a dispersive Lorentzian line was retained in the imaginary part. This can be explained by $T_{2\rho}$ relaxation that leads to a decay of coherence in the tilted frame, i.e., of magnetization in the $x''y'' \approx x'z'$-plane. Therefore,

only the component parallel to $\boldsymbol{\nu}_{\mathrm{eff}}$ which decays with the longer $T_{1\rho}$ is visible in the steady state.

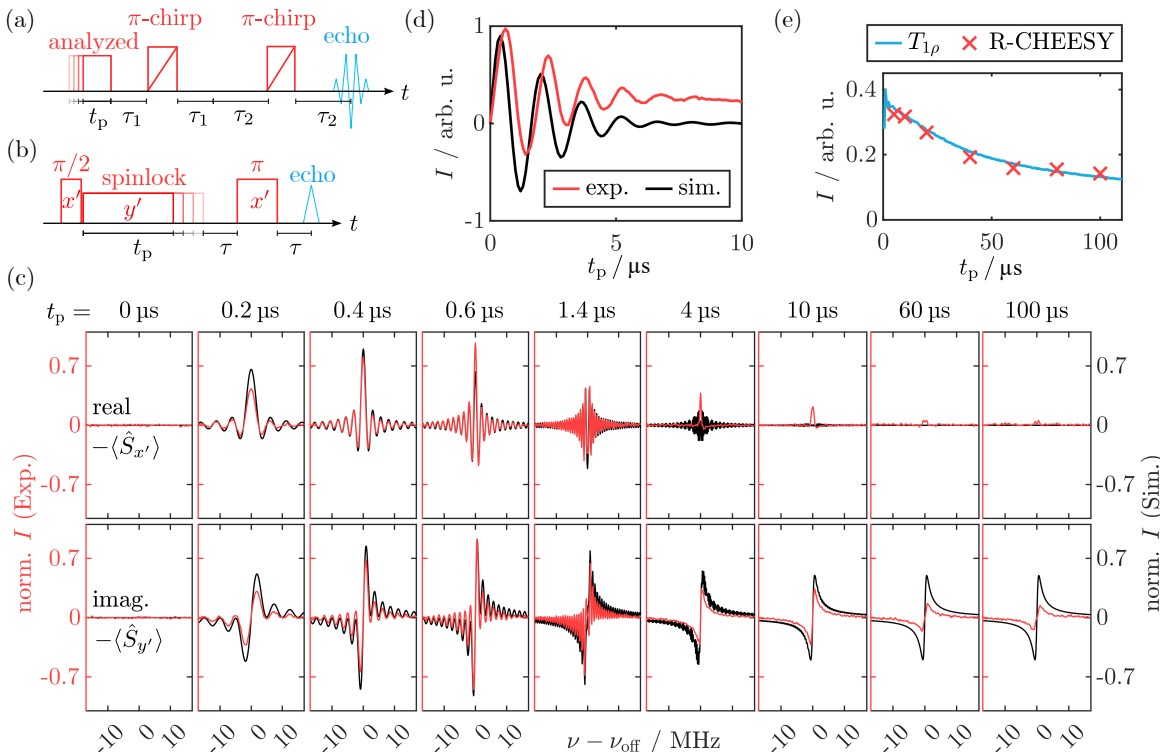

**Figure 6.** (a) R-CHEESY pulse sequence. (b) Pulse sequence for measuring longitudinal SL relaxation $T_{1\rho}$. (c) Experimental (red) and simulated (black) R-CHEESY spectra for different $t_\mathrm{p}$ with their real and imaginary component. (d) Experimental (red) and simulated (black) Rabi nutation obtained from the R-CHEESY excitation profiles for varied pulse lengths $t_\mathrm{p}$ as the real intensity at $\nu - \nu_{\mathrm{off}} = 0$. FT of the experimental data yielded $\nu_1 = 0.63\,\mathrm{MHz}$. (e) Intensity of the dispersive steady state signal from figure (c) and comparison to the $T_{1\rho}$ trace measured with $\nu_1 \approx 0.8\,\mathrm{MHz}$. For experimental parameters, see Sect. B5.

This is further validated by the decay of the experimental steady state signal along $y'$ for $t_\mathrm{p} \gtrsim 10\,\mathrm{\mu s}$. Overlay of the maximum $y'$-signal intensity as a function of $t_\mathrm{p}$ with a $T_{1\rho}$ decay measured at similar spin lock power showed good agreement (see Fig. 6 (b) and (e)). Thus, this decay could be attributed to $T_{1\rho}$ relaxation. As no longitudinal relaxation was included in our theoretical framework, this effect was not reproduced in the simulations. Simulating this effect is complicated by the fact that standard





longitudinal relaxation ultimately approaches a non-zero equilibrium distribution. How to account for this in the case of $T_{1\rho}$ is disputed in literature and goes beyond the scope of this work (e.g., Slichter (1990, p. 243), Levitt (2013, p. 316)). Development of a simulation procedure and further benchmark experiments should instead be the objective of future work.

### 4.2.3 Comparison to Bloch equations

As we assume an isolated $S = 1/2$ electron spin, in principle, one can represent the spin system by a magnetization vector in three-dimensional Cartesian space and calculate its evolution from the Bloch equations (Bloch, 1946). Being defined in the rotating frame, this is analogous to the density matrix calculations without transformation into the tilted frame. Thus, it leads to errors when implementing relaxation in the case of long MW pulses (see below). Redfield (1955) proposed a modification of the Bloch equations in the limit of strong irradiation in which the $x'y'$-relaxation was divided into a fast process perpendicular to the irradiation field and a slower one for the parallel component. Indeed, this modification resulted in line narrowing compared to the steady state solution of the standard Bloch equations under strong driving conditions (Schenzle et al., 1984; DeVoe and Brewer, 1983). Although this model is similar to our tilted frame approach, it is different in that there are now two axes with slow relaxation, and that it does not directly depend on the spin offset. In contrast to this, we analyzed the Bloch propagation in the tilted frame, in analogy to our density matrix calculations. Solution of the Bloch equations transformed in the tilted frame yielded the following rotating frame magnetization vector (for the derivation, see Sect. A14):

$$
\boldsymbol{M}'(t) = \begin{pmatrix} \sin(\omega_{\text{eff}}t)\sin(\theta)M_0 \cdot e^{-t/T_{2\rho}} \\ \left[-\cos(\omega_{\text{eff}}t)\sin(\theta)\cos(\theta)\cdot e^{-t/T_{2\rho}} + \sin(\theta)\cos(\theta)\right]M_0 \\ \left[\cos(\omega_{\text{eff}}t)\sin^2(\theta)\cdot e^{-t/T_{2\rho}} + \cos^2(\theta)\right]M_0 \end{pmatrix} \tag{16}
$$

For the limiting case of infinite microwave irradiation ($t \to \infty$), all terms including an exponential decay vanished, and an analytical steady-state solution was obtained:

$$
\lim_{t\to\infty} \boldsymbol{M}'(t) = \begin{pmatrix} 0 \\ \sin(\theta)\cos(\theta)M_0 \\ \cos^2(\theta)M_0 \end{pmatrix} = \begin{pmatrix} 0 \\ \frac{(\nu-\nu_{\text{off}})\nu_1}{\nu_1^2+(\nu-\nu_{\text{off}})^2} \\ 1 - \frac{\nu_1^2}{\nu_1^2+(\nu-\nu_{\text{off}})^2} \end{pmatrix} \tag{17}
$$

This steady-state expression was already proposed by Abragam (1961) and is in agreement with recent literature on EDNMR (Nalepa et al., 2014; Cox et al., 2017). Comparison with the experimental steady state $y'$- and $z'$-profiles showed quantitative agreement (Fig. 7 (a) and (b), compare to Fig. 4 (c)). Additionally, in contrast to the approaches by Abragam (1961) and Nalepa et al. (2014), the complete spin trajectory during a MW pulse including relaxation could be calculated analytically using Eq. 16. The results for a spin offset $\Omega_S/2\pi = 1.7\,\text{MHz}$ and $\nu_1 = 0.77\,\text{MHz}$ are shown as a black curve in Fig 7 (c) for a pulse length $t_{\text{p}} = 10\,\mu\text{s}$. Starting from $\boldsymbol{M}' = (0,0,M_0)^{\text{T}}$, the magnetization vector precesses around $\boldsymbol{\nu}_{\text{eff}}$ (gray arrow) and approaches a steady state ($y'$- and $z'$-components marked by black dashed line) that lies on $\boldsymbol{\nu}_{\text{eff}}$. The whole evolution occurs in a plane perpendicular to $\boldsymbol{\nu}_{\text{eff}}$ that cuts the $z'$-axis at $M_0$.

Again, we compared our results to propagation in the rotating frame by numerical solution of the Bloch equations in the rotating frame (for details, see Sect. A14). Analysis of this trajectory (Fig 7 (c), blue) showed that there is a decay of the





vector components parallel to the effective field in addition to the precession around $\boldsymbol{\nu}_{\text{eff}}$. This effect is already clearly visible

490    for $t_{\text{p}} = 10\,\mu\text{s}$ and culminates in $\boldsymbol{M}(t \to \infty) = \boldsymbol{0}$ for even longer pulses (not shown). In agreement with the density matrix

simulations in the rotating frame, this lead to an increased hole depth for a certain offset $\Omega_S$ and thus, to a broadening of the

burned hole. Quantitative analysis of the width of the calculated hole profiles by their full-width half-maximum (FWHM) value

(Fig. 7 (d)) showed that, while the tilted frame solution approached a steady state at the theoretical value of $2\nu_1$ (horizontal

black line), the width of the hole calculated in the rotating frame continuously increased. This effect becomes significant for

495    $t_{\text{p}} \gtrsim T_{2\rho}$ (vertical black line). The experimental FWHM (from the data in Fig. 4) shows only a slight increase of the FWHM

for long $t_{\text{p}}$ that is not consistent with the rotating frame results. We attribute the comparably small increase of the FWHM to

spectral diffusion as observed for example by Hovav et al. (2015).

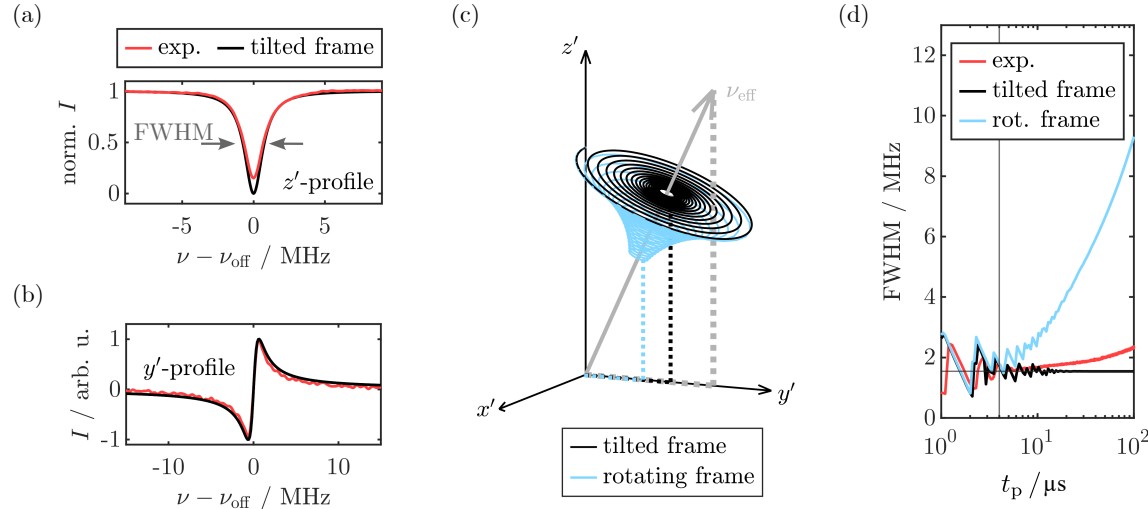

**Figure 7.** (a), (b): Experimental steady state $z'$ and $y'$ profiles ($t_{\text{p}} = 10\,\mu\text{s}$) and analytical solution from Eq. 17. Experimental MW powers

((a): $\nu_1 = 0.77\,\text{MHz}$, (b): $\nu_1 = 0.63\,\text{MHz}$) were used for calculating the analytical solution. (c) Evolution of $\boldsymbol{M}(t)$ calculated analytically

from the tilted frame solution of the Bloch equations (Eq. 16, black) and comparison to the numerical solution for the standard Bloch

equation in the rotating frame (Eq. A21, light blue). Effective MW field vector $\boldsymbol{\nu}_{\text{eff}}$ as a gray vector and its components $\nu_1 = 0.77\,\text{MHz}$ and

$\nu - \nu_{\text{off}} = 1.7\,\text{MHz}$ as dotted gray lines along $y'$ and $z'$. $y'$- and $z'$-components of $\boldsymbol{M}(t_{\text{p}})$ as black and light blue dotted lines after $t_{\text{p}} = 10\,\mu\text{s}$

for the tilted and rotating frame solution, respectively. (d) FWHM of the calculated inversion profiles using both the tilted (black) and the

rotating frame (light blue) expression as a function of $t_{\text{p}}$. The horizontal and vertical line mark the values $\text{FWMH} = 2\nu_1 = 1.54\,\text{MHz}$

and $t_{\text{p}} = T_{2\rho} = 4\,\mu\text{s}$, respectively. Oscillations for small $t_{\text{p}}$ are caused by the oscillating side lobes of the initial sinc-shaped profiles. For

additional parameters, see Sect. B6.



# 5 Conclusions

In this work, we demonstrated the feasibility of shaped pulse experiments using periodic phase modulations and chirp pulses on a commercial Bruker E-680 W-band spectrometer. These experimental tools were then applied to the analysis of electron spin dynamics during spin locking. In particular, we reported that spin dynamics during MW irradiation obey principles similar to the bare state, i.e., Larmor precession around effective fields and transverse relaxation in the eigenframe of the complete spin Hamiltonian. We performed tilted frame experiments to measure SL relaxation times, and to analyze the relation between bare and SL dynamics. Inversion and excitation profiles of low-power MW pulses were used to benchmark density matrix simulations of these SL dynamics in a reduced spin system. Quantitative agreement was reached using experimental parameters without modification. Additional *Spinach* based simulations, including electron-nuclear coupling, powder averaging, and explicit detection, suggested that our findings can be generalized to more complex spin systems and simulation routines. Comparison of these results with simulations based on the Bloch equations showed that, in both cases, transformation into the tilted eigenframe of the MW Hamiltonian is needed for pulse lengths exceeding the SL decoherence time $T_{2\rho}$ to achieve quantitative agreement with the experimental data.

Multiple questions still need to be addressed. Knowledge of the dependence of $T_{2\rho}$ on the chemical environment could increase mechanistic understanding of this relaxation process and, ultimately, enable tuning of the relaxation properties in the SL state to increase the decoherence time in pulsed EPR experiments. Furthermore, the effect of $T_{1\rho}$ was not analyzed. A similar analysis of the longitudinal SL relaxation would be interesting, if an appropriate benchmark experiment could be designed. Additionally, a detailed, quantitative analysis of the hyperfine decoupling in the SL state could be used to estimate the applicability of hyperfine spectroscopic experiments in the tilted frame.

*Code and data availability.* All experimental data, data analysis scripts, and simulation code are available for download from the Göttingen Research Open Data Repository under https://doi.org/10.25625/B11CUC.





## Appendix A: Supplementary data and analysis

### A1    Generation of shaped pulses

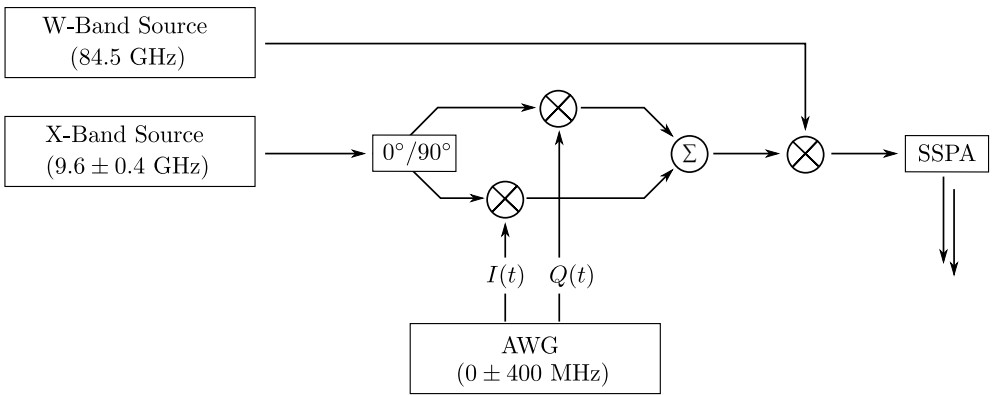

**Figure A1.** Schematic description of the pulse generation in a Bruker E680 W-band EPR spectrometer equipped with an AWG. See Sect. 3.2 for more details.

### A2    Phase stability of spectrometer

To assess the phase stability of the pulses produced by the AWG, we performed a spin echo experiment with transient detection as described by Endeward et al. (2023). The results are displayed in Fig. A2. Analysis of the data shows that strong phase noise is observable, likely due to a deteriorating performance of the $84.5\,\text{GHz}$ MW source. When looking at the average of
hundred shots (filled circles), the phase agrees reasonably well with the theoretical value (red 'x'-symbols), suggesting that phase cycling is possible whenever large numbers of scans are used. This is in agreement with our experiments where we observe efficient phase cycling. However, this procedure will lead to loss in signal to noise due to partial cancellation.



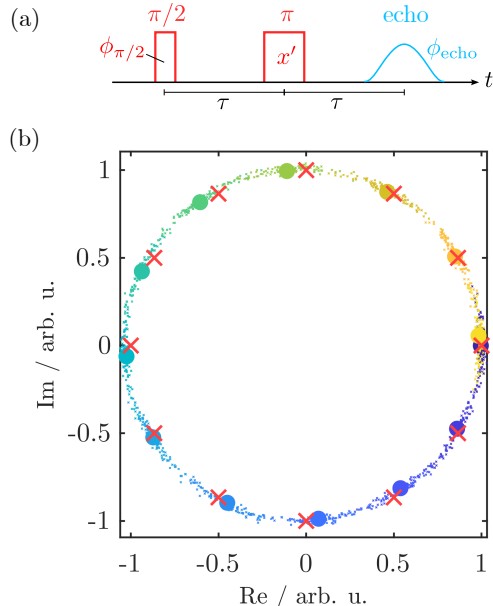

**Figure A2.** (a) Pulse sequence to measure the phase setting and stability using the AWG. The phase of the rectangular $\pi/2$-pulse was varied while the $\pi$-pulse was kept with a constant phase. (b) Real and imaginary contributions of the echo transients obtained with the pulse sequence in (a). The small dots mark hundred individual scans for each phase $\phi_{\pi/2}$ belonging to one color. The average of these results is marked by a larger, filled circle of the respective color. The red 'x' symbols mark the theoretical position. Experimental Parameters: $t_\pi = 60\,\text{ns}$, $\tau = 1\,\mu\text{s}$.

### A3 Resonator profile and amplifier characterization

Figure A3 (a) shows the $\nu_1$-profile of the Bruker EN600-1021H ENDOR resonator measured with BDPA at a temperature of
$50\,\text{K}$. The profile is clearly asymmetric and shows some oscillations, which are indicative of reflections in the resonator (Doll and Jeschke, 2014; Endeward et al., 2023).

Additionally, Rabi nutation experiments were performed at different AWG output voltages to measure the amplification curve of the SSPA. This was achieved by variation of the *amplitude* parameter (Amp) in the Xepr software that scales the input $IQ$ voltage values by a factor between $0$ and $100\,\%$. Due to saturation effects, the amplification curve was clearly non-linear.
Therefore, all nominal MW powers were corrected with this curve in order to yield the target $\nu_1$.



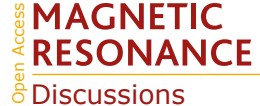

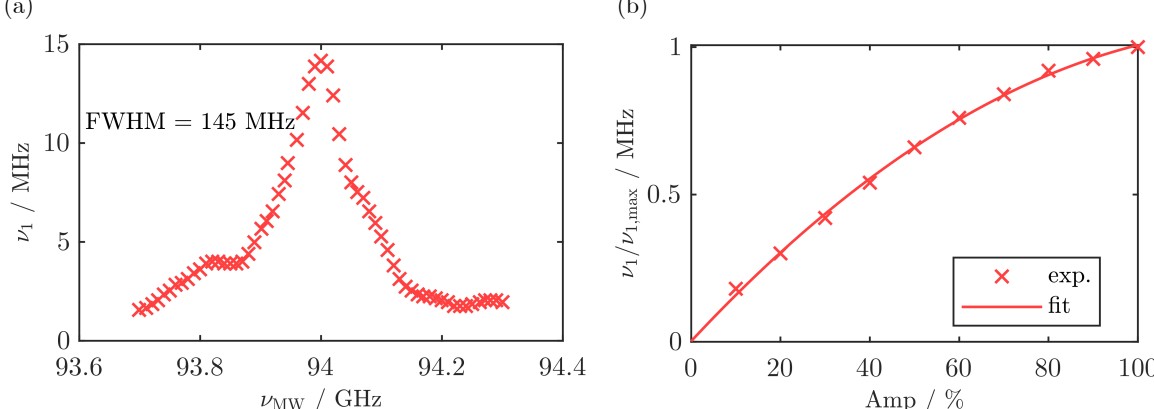

**Figure A3.** (a) Resonator profile used for the correction of the Chirp pulses. Acquisition was done at $50\,\mathrm{K}$ by measuring two-pulse Rabi nutations of BDPA at different local oscillator frequencies. (b) Amplification curve of the solid state amplifier used for correction of the Chirp pulses. The data was obtained by room temperature measurements of BDPA Rabi nutations at different attenuation values of the AWG output voltage. The latter are given as a relative *amplitude* (Amp) values that can be between 0 and $100\,\%$. Pulse Sequence: $\mathrm{pulse} - T - \pi/2 - \tau - \pi - \tau - \mathrm{echo}$. Experimental Parameters: $t_\pi \approx 40 - 400\,\mathrm{ns}$, $T \approx 10\,\mu\mathrm{s}$, $\tau \approx 0.3 - 1\,\mu\mathrm{s}$.

## A4 Generation of PM pulses

PM pulses were generated with a home-written *MALTAB* routine following Eq. 5 (Wili et al., 2020). Figure A4 (a) shows two PM pulses with phases of $\phi_{\mathrm{PM1}} = 0$ and $\phi_{\mathrm{PM2}} = \pi/2$, respectively. The $I(t)$ and $Q(t)$ values are calculated from $\phi_{\mathrm{MW}}$ using Eq. A1

$$I(t) = \sin(\phi_{\mathrm{PM}}); \qquad Q(t) = \cos(\phi_{\mathrm{PM}}) \tag{A1}$$

A representative pulse shape including two PM pulses is shown in Fig. A4 (b). Note that some distortions caused by the comparably large time increment of $10\,\mathrm{ns}$ are visible for $\nu_{\mathrm{PM}} = 15\,\mathrm{MHz}$. However, choosing a smaller time increment was impossible due to the limited number of $IQ$-values for a single pulse shape supported by the *Xepr* software.





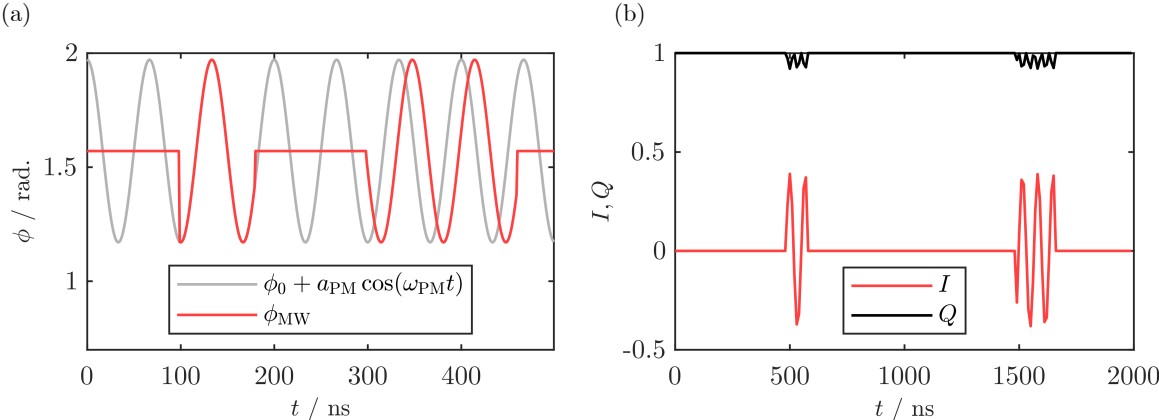

**Figure A4.** (a) MW phase $\phi_{\mathrm{MW}}$ during a representative pulse shape including PM pulses (red) and corresponding carrier wave (after Wili et al., 2020). The following parameters were used: $\nu_1 = \nu_{\mathrm{PM}} = 15\,\mathrm{MHz}$, $\phi_0 = \pi/2$, $a_{\mathrm{PM}} = 0.4$, $\tau_0 = 100\,\mathrm{ns}$, $\tau_1 = \tau_2 = 200\,\mathrm{ns}$, $t_{\mathrm{PM1}} = 80\,\mathrm{ns}$, $t_{\mathrm{PM2}} = 160\,\mathrm{ns}$, $\phi_{\mathrm{PM1}} = 0$, $\phi_{\mathrm{PM2}} = \pi/2$. A time increment of $1\,\mathrm{ns}$ was used for better visualization. Note the shift of PM2 with respect to the carrier wave due to its PM-phase. (b) $I(t)$ and $Q(t)$ values of a representative pulse shape including PM pulses as uploaded to the spectrometer. The following parameters were used: $\nu_1 = \nu_{\mathrm{PM}} = 15\,\mathrm{MHz}$, $\phi_0 = \pi/2$, $a_{\mathrm{PM}} = 0.4$, $\tau_0 = \tau_2 = 500\,\mathrm{ns}$, $\tau_1 = 1000\,\mathrm{ns}$, $t_{\mathrm{PM1}} = 80\,\mathrm{ns}$, $t_{\mathrm{PM2}} = 160\,\mathrm{ns}$, $\phi_{\mathrm{PM1}} = 0$, $\phi_{\mathrm{PM2}} = \pi/2$. A time increment of $10\,\mathrm{ns}$ was used in agreement with the experiments.

## A5 Calibration of $\nu_{\mathrm{PM}}$ and measurement of the $\nu_1$ profile

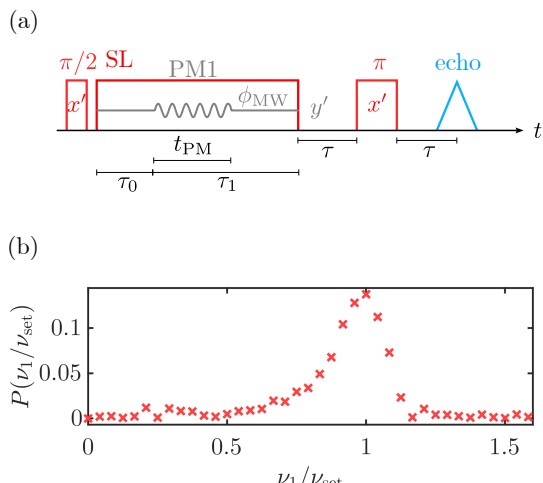

**Figure A5.** (a) Pulse sequence for measuring the frequency-stepped SL resonance spectrum and result for $\nu_{\mathrm{set}} = 12\,\mathrm{MHz}$ (b). *Experimental Parameters:* $t_\pi = 48\,\mathrm{ns}$, $\tau_0 = 1\,\mu\mathrm{s}$, $\tau_1 = 2\,\mu\mathrm{s}$, $t_{\mathrm{PM}} = 1\,\mu\mathrm{s}$, $a_{\mathrm{PM}} = 0.04$, $\tau = 0.3\,\mu\mathrm{s}$, SPP $= 5$, 4-step PC of $\pi$ detection pulse.





The frequency-stepped SL resonance spectrum was measured to set $\nu_{\text{PM}} = \nu_1$ (Wili et al., 2020). In this experiment, a single long PM pulse with low modulation amplitude ($t_{\text{PM}} = 1\,\mu s$ and $a_{\text{PM}} = 0.04$) was applied during SL (Fig. A5 (a)). This caused a signal decrease proportional to the degree of matching of $\nu_1$ and $\nu_{\text{PM}}$. The frequency $\nu_{\text{PM}}$ with a maximal signal decrease was used in the PM pulse experiments (i.e., $\nu_{\text{set}}$).

To obtain a discrete probability distribution function of $\nu_1$ (i.e., $P(\nu_1)$) from this experiment, the spectrum was vertically
shifted and normalized to a sum of one. The abscissa was scaled by $\nu_{\text{set}}$ to allow transfer between experiments with different $\nu_{\text{set}}$ (Fig. A5 (b)). For simulation of MW inhomogeneity, all data points exceeding $0.3P(\nu_1/\nu_{\text{set}} = 1)$ were selected. Calculations were repeated for each of these $\nu_1/\nu_{\text{set}}$ values and the results weighted by $P(\nu_1/\nu_{\text{set}})$. For the simulation of SL echoes, linear interpolation was used between the selected data points to increase the resolution of the $\nu_1/\nu_{\text{set}}$-axis.

## A6 Generation of chirp pulses

Shapes of chirp pulses were generated using a home-written *MATLAB* routine. After defining the pulse length $t_{\text{p}}$ and sweep width $\nu_{\text{SW}}$, an incremented frequency array of $\nu_{\text{AWG}}$ was generated that spanned the whole sweep range. For each increment $i$, the length $\Delta t(i)$ was adapted using the experimental resonator profile in Fig. A3 (a) so that the critical adiabaticity $Q_{\text{crit}}(i)$ (Eq. A2) was constant throughout the pulse and equal to its mean value $\overline{Q_{\text{crit}}}$ (Baum et al., 1985; Jeschke et al., 2015).

$$Q_{\text{crit}}(i) = \frac{2\pi[\nu_1(i)]^2 \Delta t(i)}{\Delta \nu} \overset{!}{=} \overline{Q_{\text{crit}}} \tag{A2}$$

$$\overline{Q_{\text{crit}}} = \frac{2\pi \overline{\nu_1^2} t_{\text{p}}}{\nu_{\text{SW}}} \tag{A3}$$

Here, $\nu_1(i)$ is the Rabi frequency for increment $i$ as obtained from the resonator profile. $\Delta \nu$ corresponds to the constant frequency increment $\nu_{\text{AWG}}(i+1) - \nu_{\text{AWG}}(i)$. $\Delta t_{\text{p}}$ is the length of time increment $i$ and $\overline{\nu_1^2}$ is the mean value of $\nu_1^2$ as obtained from the resonator profile.

The non-linear frequency sweep was interpolated to fit the constant AWG sampling rate of $\Delta t_{\text{AWG}} = 0.625\,\text{ns}$ and converted
to a phase modulation by Eq. A4 (Doll and Jeschke, 2017).

$$\phi_{\text{AWG}}(t) = \int_0^t 2\pi \nu_{\text{AWG}}(t')\text{d}t' + \phi_{\text{AWG}}(0) \tag{A4}$$

This phase was inserted into Eq. A1, yielding the AWG input $I(t)$ and $Q(t)$. A wideband, uniform rate, smooth truncation (WURST) amplitude modulation function was applied to these $I(t)$ and $Q(t)$ values in order to reduce pulse edge artifacts (Kupce and Freeman, 1995), e.g.:

$$I_{\text{WURST}}(t) = I(t) \cdot \left(1 - \left|\cos\left(\frac{\pi t}{t_{\text{p}}}\right)\right|^{n_{\text{w}}}\right) \qquad \text{where } t \in [0, t_{\text{p}}] \tag{A5}$$

$n_{\text{W}}$ is a parameter that reflects the steepness of the WURST modulation function. For the experiments shown in this work $n_{\text{W}} = 50$ was used. The final pulse shapes were obtained after correction with the amplifier profile (see Fig. A3 (b)).

In order to assess the performance of these pulses, we measured inversion profiles of representative pulse shapes. We used a spin echo experiment as illustrated in Fig. A6 (a,b) and compared the echo intensity with and without the pulse of interest (Doll





and Jeschke, 2014). In contrast to the work by Bahrenberg et al. (2017), we varied the frequency of the detection sequence and the $B_0$ field, while keeping the analyzed pulse at a constant frequency centered at the resonator dip with $\nu_{\mathrm{resonator}} = \nu_{\mathrm{LO}}$. In this way, the shaped pulses stayed fixed with respect to the resonator profile and we could monitor the effect of the resonator bandwidth on the inversion properties of the pulses. In the case where the detection frequency is fixed at the position of the resonator dip, we would expect overestimation of the pulse performance because inversion is always measured at the position with maximal $\nu_1$. With this procedure, two factors had to be taken into account. First, $\nu_1$ of the detection pulses depends on

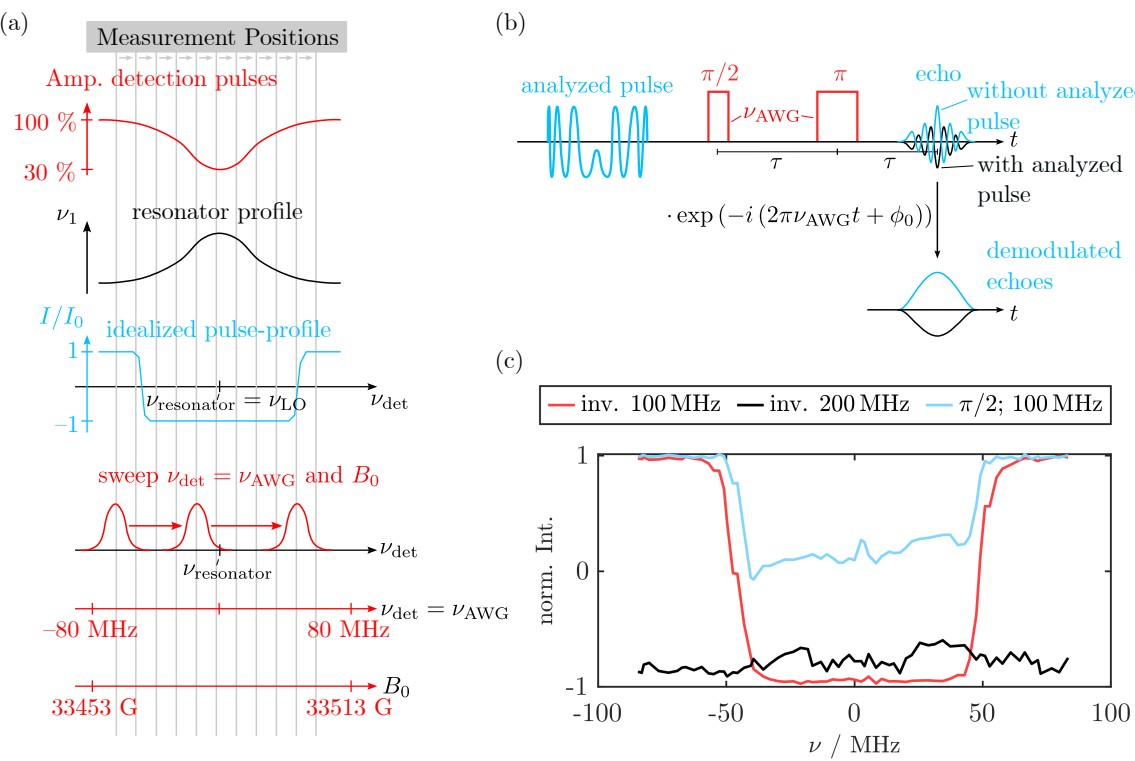

**Figure A6.** (a) Experiment used for acquisition of inversion profiles together with the used pulse sequence (b), following the procedure of Doll and Jeschke (2014). (c) Representative inversion profiles of $1\,\mu s$ WURST chirp pulses corrected for the resonator profile. Inversion (inv.) pulses were performed with maximal $\nu_1$ while the $\nu_1$ value of the $\pi/2$-pulse was optimized with a chirp nutation (see Fig. A7). The frequency value in the legend gives the sweep range the pulses were optimized for. Experimental Parameters: $t_\pi = 72\,\mathrm{ns}$, $\tau = 0.6\,\mu s$, SPP = 5.


their position with respect to the resonator profile. In order to keep the pulse length constant, the AWG output voltage (set by the *amplitude* parameter in the Xepr software) of the detection pulses was scaled with the inverse resonator profile (Doll and Jeschke, 2014). Second, signal detection by the spectrometer is done in a quadrature detection scheme by demodulation with $\nu_{\mathrm{PLO}}$ and $\nu_{\mathrm{x}}$. Thus, the resulting echo was oscillating with $\nu_{\mathrm{AWG}}$ that was different for each data point. We demodulated the

echoes using a complex exponential function oscillating with $\nu_{\mathrm{AWG}}$ (see Fig. A6 (b)). $\phi_0$ was optimized for the background





spectrum without the analyzed pulse and then used for the echo with pulse without further modification. Figure A6 (c) shows the inversion profiles of representative Chirp pulses with a pulse length of $t_\text{p} = 1\,\mu\text{s}$ in a frequency range of $\pm 80\,\text{MHz}$. While the $\pi$-pulse with $\nu_\text{SW} = 100\,\text{MHz}$ shows good inversion efficiency over the whole sweep range, some artifacts are visible in the case of $\nu_\text{SW} = 200\,\text{MHz}$. The pulse with $\nu_1$ optimized for a $\pi/2$-pulse shows a reasonably flat profile with $I/I_0 \approx 0$ as expected.


### A7 Characterization of chirp echoes

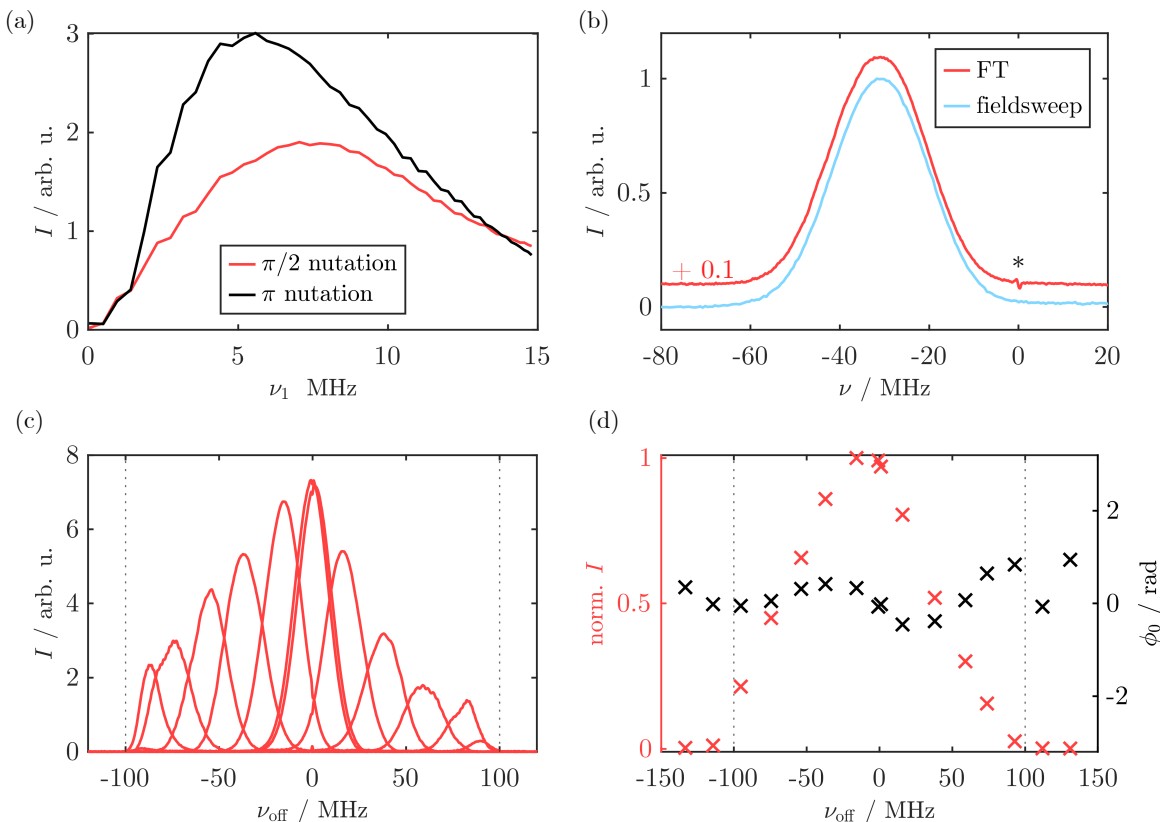

**Figure A7.** (a) Representative chirp echo nutations for the $\pi/2$ and $\pi$ chirp pulses using a $1\,\mu\text{s}$, $0.5\,\mu\text{s}$ chirp echo with a sweep width of $200\,\text{MHz}$. The $\pi/2$- nutation was performed with $\nu_{1,\pi} = 10\,\text{MHz}$ while the $\pi$ nutation was acquired using the optimized value of $\nu_{1,\pi/2} = 7.7\,\text{MHz}$. (b) Comparison of the BDPA EPR spectrum obtained by FT of a Chirp echo (red) and by echo-detected EPR (light blue). The asterisks denotes a zero-frequency artifact in the FT. (c) FT EPR spectra of BDPA as a function of the frequency offset. Acquisition was done by variation of the magnetic field. (d) Integral of each FT spectrum as a function of the frequency offset (red) as well as its corresponding phase (black). The dashed lines mark the sweep range of the chirp pulses. Note that the center of the integral profile in (d) is shifted to negative $\nu_\text{off}$ values by approximately $10\,\text{MHz}$, which was presumably caused by tuning imperfections. Pulse Sequence: $\pi/2_\text{chirp} - \tau - \pi_\text{chirp} - \tau - \text{echo}$. Experimental Parameters: $t_{\pi/2} = 1\,\mu\text{s}$, $t_\pi = 0.5\,\mu\text{s}$, $\tau \approx 1 - 2\,\mu\text{s}$ SPP $\approx 20$, 8-step PC.





Figure A7 (a) shows the chirp nutations to determine the ideal $\nu_1$ value for both the $\pi/2$ and the $\pi$ chirp pulse (Doll and Jeschke, 2014). Despite the low $Q_{\text{crit}}$ of the chirp $\pi$ pulse (see Sect. 3.5), the FT of a chirp echo of BDPA showed quantitative agreement with the corresponding ED-EPR spectrum, as it is shown in Fig. A7 (b). For the latter, the field axis $B$ was converted

into a frequency axis $\nu$ using $\nu = \nu_{\text{off,FS}} - g_e \mu_B h B$. $\nu_{\text{off,FS}}$ was selected manually to align the ED-EPR with the FT spectrum. Figure A7 (c) shows the phased chirp echo FT spectrum of BDPA as a function of the offset relative to the center of the resonator profile (Endeward et al., 2023). The corresponding integral and the phase $\phi_0$ of the signal are shown in (d). The intensity profile closely follows the resonator profile.

## A8   SL Rabi nutations

The signals observed in the FT of Fig. 1, centered at the SL Rabi frequency $\frac{a_{\text{PM}}\nu_1}{2}$ and at $2\nu_1$, were also retained when varying $\nu_1 = \nu_{\text{PM}}$ (see Fig. A8). Again, simulations agreed well with the experimental data and the theoretical signals. If a single bare

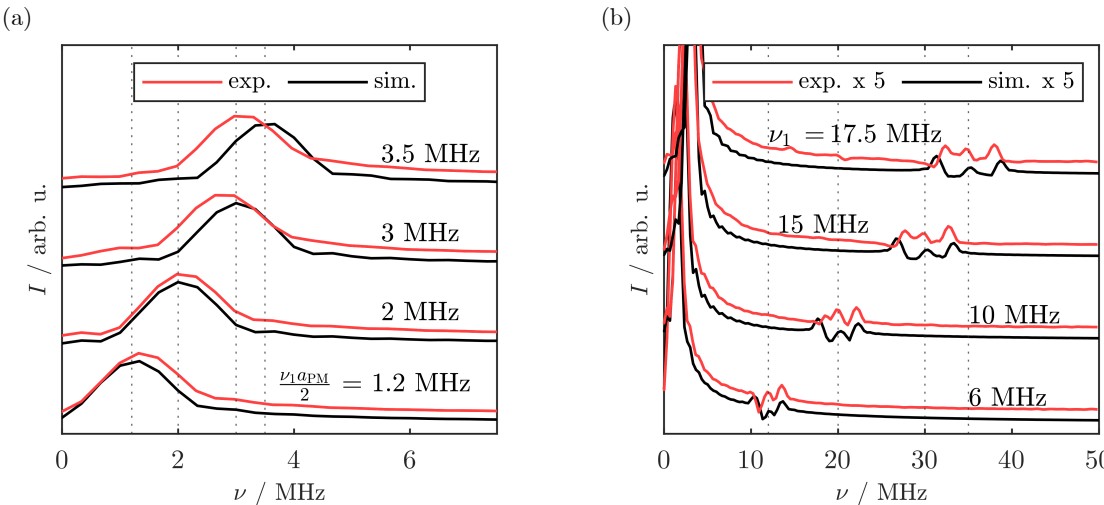

**Figure A8.** FT spectra of the experimental SL Rabi nutation traces (red) obtained with a fixed $a_{\text{PM}} = 0.4$ and different $\nu_1$. (a) low frequency region in the order of the SL Rabi frequency $\frac{a_{\text{PM}}\nu_1}{2}$. (b) Complete frequency region. FTs of simulated SL Rabi nutations with the same MW power are displayed in black. Vertical lines mark the positions where a signal is expected, i.e., at $\nu = 2\nu_1$ in (a) and at $\nu = \frac{a_{\text{PM}}\nu_1}{2}$ in (b). For experimental parameters, see Sect. B1.

spin offset $\Omega_S$ was introduced to the simulation, a strong distortion of the FT spectra occurred for $\Omega_1 \gtrsim \omega_1$ (Fig. A9 (a)). In contrast, simulation of a Gaussian distribution of $\Omega_S$-values with FWHM $= 24\,\text{MHz}$ did not yield these artifacts (Fig. A9 (b)).

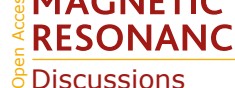

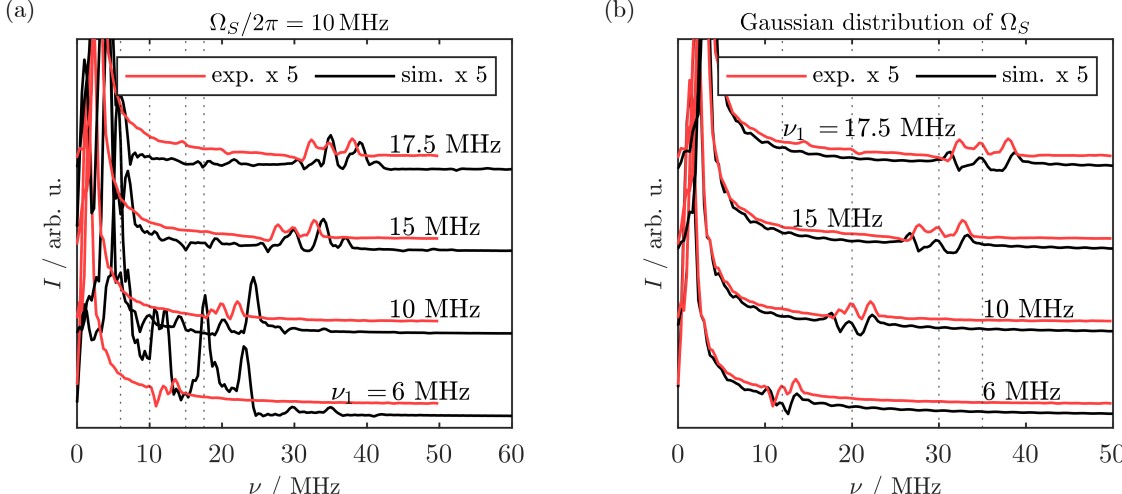

**Figure A9.** FTs of SL Rabi nutations (red, as in Fig. A8) compared to simulations with a single electron spin offset $\Omega_S/2\pi = 10\,\text{MHz}$ (a) and with a Gaussian distribution of $\Omega_1$-values (b). For the latter, the weighted sum of 2001 traces calculated for $\Omega_S/2\pi$ between $-24\,\text{MHz}$ and $24\,\text{MHz}$ was calculated using weighting factors from a Gaussian distribution function centered at $0\,\text{MHz}$ and with $\text{FWHM} = 24\,\text{MHz}$.

## A9   Phase of SL echo experiments

During a SL echo experiment, we can distinguish between two kinds of phase information. On the one hand, the final integrated echo has a phase depending on the magnetization phase at the end of the SL pulse. On the other hand, the SL echo itself (i.e., the one generated by the PM pulses) has a phase relative to the read-out PM3 pulse. As both of these phases carry additional experimental information, these will be discussed in the following section.

The imaginary signal component of the SL echo experiments in Fig. 2, which refers to the phase of the integrated echo, is shown in Fig. A10. Comparison of real and imaginary component shows that the signals are clearly separated in the $\tau_2$ dimension. While the real signals can be assigned to spins that are refocused at the end of $\tau_2$, the imaginary signals all stem from spins refocused at the end of $\tau_3$. Namely, (i), (ii) and (iii) can all be explained by stimulated echoes resulting from different combinations of delays. Signal (iv) is a combination of a refocused echo followed by a stimulated echo . Please note that only signals where PM3 acts as a $\pi/2$-pulse are visible because of the 2-step PC of PM3 in all experiments. The signal in (a) that is marked with an asterisk is a contribution from the real part caused by phase drifts between the experiments.

This phase separation of the SL echo signals differs from the phase of the SL echoes at the position of PM3. In order to measure this phase, quadrature detection was applied on the read-out pulse PM3. Namely, its two step phase cycle that was applied in all SL echo experiments was shifted by $90°$ so that, instead of $\phi_{\text{PM3}} = [0°, 180°]$, $\phi_{\text{PM3}} = [90°, 270°]$ was used. The results for the different phases are displayed in Fig. A11. Although the experiment was performed under nominally same conditions as the one shown in Fig. 2 (d), the relative phase of the signals with $\phi_{\text{PM3}} = [0°, 180°]$ is different in this spectrum. As the first peak is caused by a refocused echo and the second by a stimulated echo, an inverted sign would be expected for

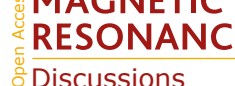

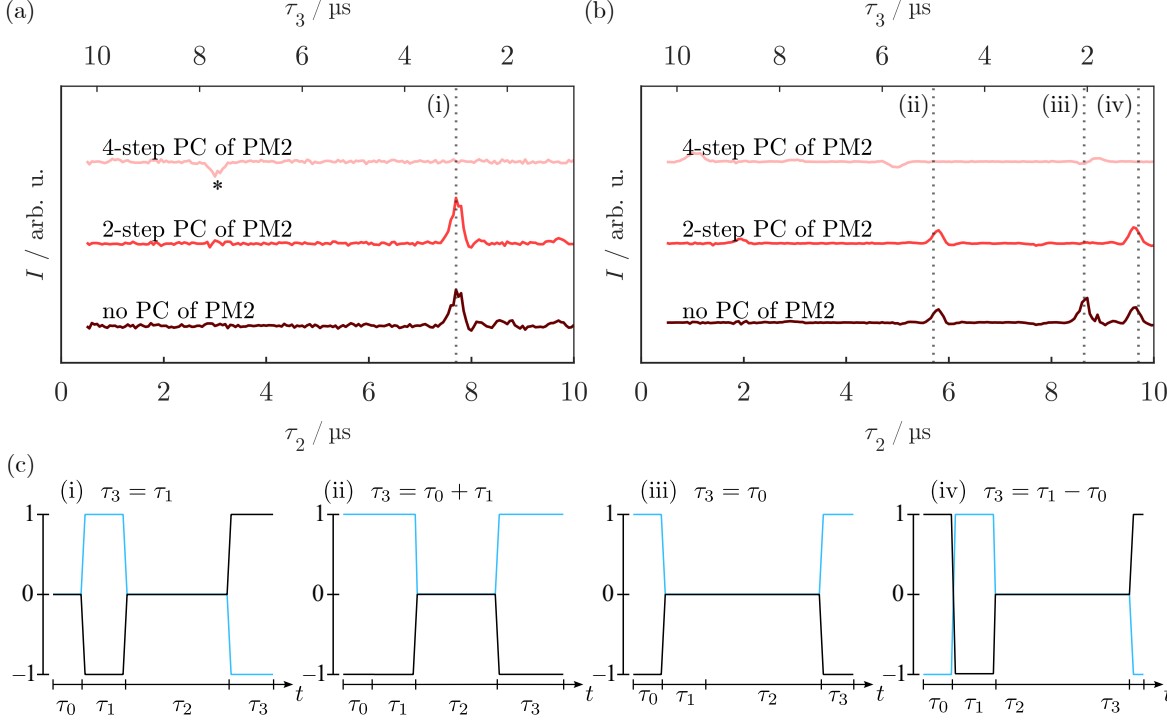

**Figure A10.** (a), (b) Imaginary signal of the SL echo experiments shown in Fig. 2 (c) and (d) for the SL echo and WOP SL echo, respectively. The asterisk marks an artifact arising from phase drifts between the experiment. (c) Coherence diagrams explaining the signals (i) to (iv). All spectra were obtained with an additional 2-step PC on PM3. Thus, only signals where PM3 acted as a $\frac{\pi}{2}$ pulse are visible. For experimental parameters, see Sect. B2.

the two, which is the case in Fig. A11 but not in Fig. 2 (d). This already shows that the phase of the echoes is not stable in our experiment. Additionally, the imaginary signal of the quadrature detection experiment (i.e., $\phi_{\text{PM3}} = [90°, 170°]$) shows significant absorptive contributions for some of the signals. This again demonstrates that the observed phase in this experiment

deviates from the theoretically expected one.



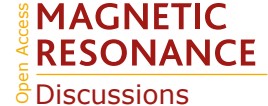

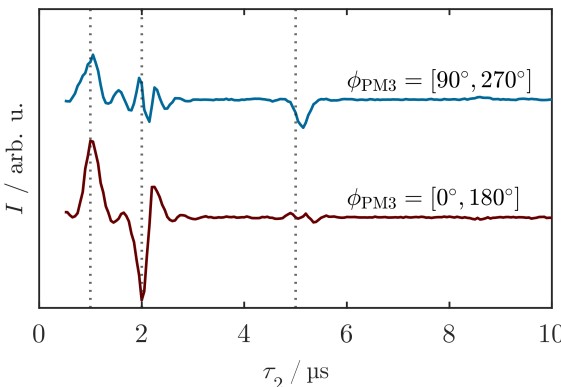

**Figure A11.** WOP SL echo measured with standard (dark red) and $90°$ phase shifted (blue) PM3 pulse. The former is obtained under identical conditions as the bottom trace in Fig. 2 (d). For experimental parameters, see Sect. B2.

## A10  $T_{2\rho}$ Measurements

The SL relaxation decays and their mono- and stretched exponential least-square fits (Eq. A6) corresponding to the $T_{2\rho}$ values in Fig. 3 are shown in Fig. A12 (a) and (b). Eq. A6 and A7 were used for the fitting:

$$I = I_0 \cdot \exp\left(-\frac{\tau_1 + \tau_2}{T_{2\rho}}\right) \tag{A6}$$

$$I = I_0 \cdot \exp\left(-\left(\frac{\tau_1 + \tau_2}{T_{2\rho}}\right)^{\xi}\right) \tag{A7}$$

Oscillations are visible in the beginning of some of the experimental traces. Following literature concerned with the fitting of $T_\mathrm{m}$-traces that include electron spin echo envelope modulation (ESEEM) (Soetbeer et al., 2021), we fitted the envelope function of the oscillation maxima. Specifically, we generated a time window with a length of the inverse of the main oscillation frequency for each data point. The data point with maximal intensity was selected for each of these windows. After removing

the redundancies in this selected data set, only the data points at a maximum of the dominant oscillation were obtained. A detailed analysis of the oscillation frequencies can be found in section A11.

Statistical analysis was performed using the bootstrapping method (e.g. Press et al., 1992). For each $N$ point data set, $N_\mathrm{boot} = 1000$ synthetic data sets were generated by randomly taking $N$ points from the data set with replacement and thus, allowing points to be selected multiple times. Each of these bootstrap data sets was fitted and the final result and its error

were obtained as the mean and standard deviation of the fit parameters from all synthetic data sets. Errors are given as the $95\%$ confidence intervals (i.e., as $2\sigma$). Note that this only takes into account statistical errors and cannot give an estimate of systematic errors introduced by the fit model. These can instead be analyzed using the residuals of the fitting (Fig. A12 (c)). Note that the oscillations in the residuals for small $\tau_1 + \tau_2$ indicate that the removal of oscillating components in the decay traces is incomplete (see Sect. A11 for more detail). Analysis of the Pearson correlation coefficient for $T_{2\rho}$ and $\xi$ in the bootstrap



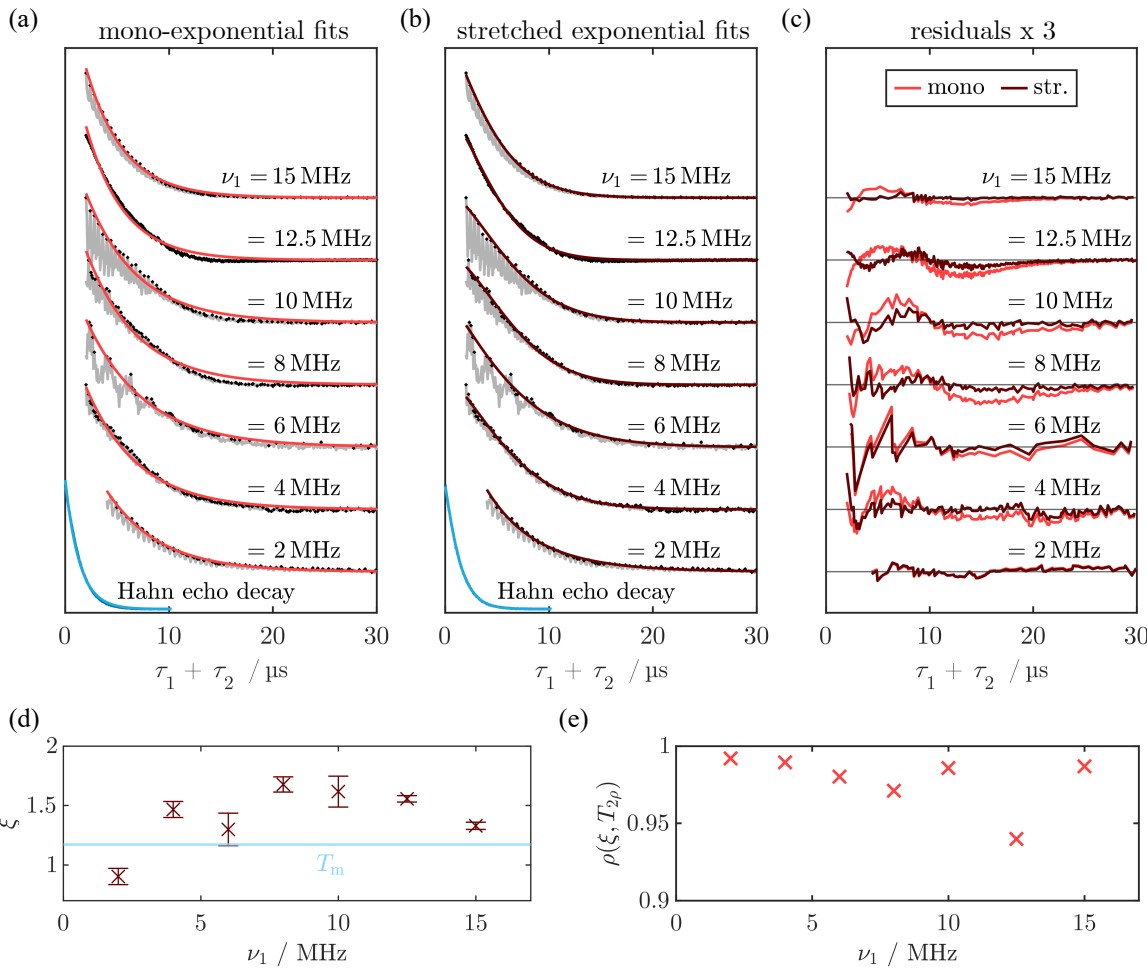

**Figure A12.** Experimental SL echo decay traces (grey) and the corresponding mono-exponential (a) and stretched exponential (b) fits. Data points selected for the envelope fitting are displayed as black dots. The corresponding $T_{2\rho}$ values can be found in Fig. 3. (c) Residuals of both fits, scaled by a factor of three for better visualization. (d) Stretching exponents $\xi$ for the $T_{2\rho}$ fits (dark red) and for $T_\mathrm{m}$ (blue horizontal line) in (b). (e) Pearson correlation coefficient for $\xi$ and $T_{2\rho}$ of the different fits in (b) calculated from the fitting results of the bootstrap samples. For experimental parameters, see Sect. B3.

samples was performed using the *corrcoef* command in *MATLAB* (see Fig.A12 (e)). For all values of $\nu_1$, the Pearson correlation of the fitting parameters $\rho(\xi, T_{2\rho})$ was above 0.9 which shows that the parameters are highly correlated.

## A11   Oscillations in $T_{2\rho}$ Traces

Distortions of the SL state relaxation traces, consisting of oscillations for small $\tau_1 + \tau_2$ values, were observed in most tilted frame echo decay experiments. As deuteration of the used BDPA radical showed no effect on the oscillation frequency (data





not shown), ESEEM-like modulation caused by residual hyperfine couplings could be excluded. Moreover, a phase shift of the oscillation by $\pi$ was observed when changing $\phi_{\mathrm{PM}}$ by $\frac{\pi}{2}$ for all three PM pulses simultaneously (see Fig A13 (a)), which suggested an effect depending on an oscillation with $|\cos(\phi_{\mathrm{PM}})|$. FT of the experimental $T_{2\rho}$ decays for different $\nu_1$ and $\tau_1 + \tau_2$ increments (red in Fig. A13 (c) and (d)) showed seemingly random but well defined oscillation frequencies.

Considering the principle of PM pulses (Fig. A13 (b), (Wili et al., 2020)), it is apparent that, depending on the time $t$
between the beginning of the spin lock and the PM pulse and on the PM phase $\phi_{\mathrm{PM}}$, there is a jump in $\phi_{\mathrm{MW}}$ at the beginning and end of PM pulses (blue and black x-symbols in Fig. A13 (b)). Its magnitude depends on $|\cos(\nu_{\mathrm{PM}}t + \phi_{\mathrm{PM}})|$. As such abrupt changes in pulses are known to produce artifacts, these were a plausible explanation for the observed deviations from an ideal echo decay. Because the $80\,\mathrm{ns}$ increment of $\tau_1 + \tau_2$ in our experiments was larger than the Nyquist threshold of half of the inverse PM frequency $\frac{1}{2\nu_{\mathrm{PM}}} \approx 30\,\mathrm{ns}$, not the $\nu_{\mathrm{PM}}$ oscillation but a low frequency alias would be detected in this case (see
Fig. A13 (e)). Thus, we simulated the oscillations by selecting values from $|\cos(\omega_1 t)|$ with the experimental time increment and subsequently Fourier transformed this under-sampled oscillation (black in Fig. A13 (c) and (d)). The frequencies obtained from this FT agreed quantitatively with some of the signals observed in the experimental FT spectra of the decay traces. Additional signals could be reproduced when repeating the procedure with a $\left|\cos\left(\frac{\omega_1}{2}t\right)\right|$ function (blue in Fig. A13 (c) and (d)). This is reasonable because, while the position of PM3 shifted with $\tau_1 + \tau_2$, the position of PM2 varied with $\frac{\tau_1 + \tau_2}{2}$. Hence,
a combination of the pulse edges caused by both PM2 and PM3 could explain most of the observed signals. Importantly, this method also predicts that there is no oscillation in the trace for $\nu_1 = 12.5\,\mathrm{MHz}$ which agrees well with the experiment. In the future, we could detect the decay traces with a time increment $\tau_1 + \tau_2$ adjusted so that no oscillations are present in the traces. This might improve the results of the fitting for various $\nu_1$.



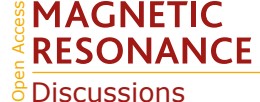

**Figure A13.** (a) Excerpt of $T_{2\rho}$ nutation traces obtained with $\nu_1 = 15\,\text{MHz}$ and $\phi_{\text{PM}} = 0, \frac{\pi}{2}$. (b) Schematic drawing of arbitrary PM pulses for $\phi_{\text{PM}} = 0, \frac{\pi}{2}$ (black, blue). The first points of the PM pulses are marked with an x and a continuous cosine wave as a reference for $\phi_{\text{PM}} = 0$ is marked in gray. FTs of the experimental $T_{2\rho}$-traces for different MW powers $\nu_1$ (c) and increments of $t = \tau_1 + \tau_2$ (d) show distinct frequencies (red). Assuming undersampling of oscillations $|\cos(\omega_{\text{PM}}t)|$ and $\left|\cos\left(\frac{\omega_1}{2}t\right)\right|$ the FT of the resulting traces yielded alias frequencies in quantitative agreement with the oscillations observed in the experiment (blue and black in (c) and (d)). (e) Illustration of alias frequencies for a $\left|\cos\left(\frac{\omega_1}{2}t\right)\right|$ oscillation with $\nu_{\text{PM}} = 15\,\text{MHz}$ and different sampling increments as used in (d). For experimental parameters, see Sect. B3.





## A12 Simulation of MW inhomogeneity

Figure A14 shows the experimental Rabi nutation from the CHEESY $z'$-profiles (see Fig 4 (c)) in red. Additionally, the simulations with and without MW inhomogeneity (black and blue, respectively) are shown. Comparison with an experimental $T_{2\rho}$ decay (gray) shows that experiment and simulation including the $\nu_1$ distribution agree well and decay significantly faster than the $T_{2\rho}$ trace. In contrast, the simulation without MW inhomogeneity decays slower and closely resembles the $T_{2\rho}$ decay. The $\nu_1$ distribution for calculation of the black curve is taken from the frequency-stepped tilted frame resonance spectrum without 675 further modification.

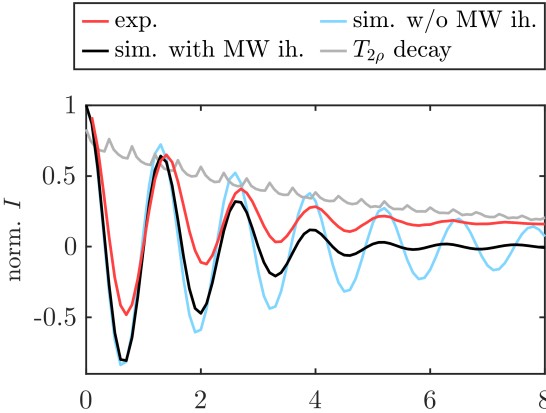

**Figure A14.** Experimental Rabi nutation obtained from the CHEESY $z$-profiles (red) and comparison with the simulation with (black) and without (blue) including the $\nu_1$ distribution function $P(\nu_1)$. For both simulations, $T_{2\rho}$ was set to $4\,\mu\text{s}$. For comparison, the experimental $T_{2\rho}$ decay ($\omega_{\text{PM}} = 15\,\text{MHz}$) is shown in gray. For experimental parameters, see Sect. B4.

## A13 Spinach simulation of the CHEESY $z$-profiles

We chose an electron spin with a rhombic $g$-tensor (see Fig. A16 (b)) as a model for our Spinach simulations which differs from the almost isotropic $g$-tensor of the BDPA radical. Setting the spin parameters to the literature values known for BDPA could not reproduce the experimental Gaussian line shape because this shape is caused by the coupling to a large number of 680 nearly equivalent protons in the BDPA molecule. Calculating the line shape accurately would require including all these nuclei into the simulations, which is computationally unfeasible. Hence, an artificial $g$-anisotropy was introduced to generate a broad enough EPR line to monitor the whole inversion profiles. However, this caused a line shape different from a Gaussian EPR line. Namely, without any coupled nuclei, this lead to a distinct rhombic powder pattern with a singularity at $\nu - \nu_{\text{off}} = 0$ (black curve in Fig. A16 (a)). This caused small artifacts in the CHEESY simulations, as the CHEESY intensity scales with the EPR 685 intensity, leading to profiles slightly too narrow in the center.





By repeating the simulations with different $T_{2\rho}$ times, the sensitivity of the simulations with regard to the relaxation time was analyzed (Fig. A15). Using the bare state transverse relaxation time $T_m = 1.4\,\mu s$ (light red) did not reproduce the experimental profiles as the decay of the sinc oscillations overestimated. This is most clearly visible for $t_p = 1.1\,\mu s$ and $1.4\,\mu s$ where significant features of the experimental profiles are missing. In contrast, the visual difference between $T_{2\rho} = 4\,\mu s$ and $10\,\mu s$ is small.

Therefore, deriving $T_{2\rho}$ solely from the CHEESY profiles is not easily possible. Importantly, omitting $T_{2\rho}$ completely could not reproduce the experimental data (light blue) because no steady state was reached (compare $t_p = 10\,\mu s$, $50\,\mu s$ and $100\,\mu s$). Including nuclei led to distinct changes in the simulated EPR spectra (Fig. A16 (a)). Namely, coupling to one proton nucleus

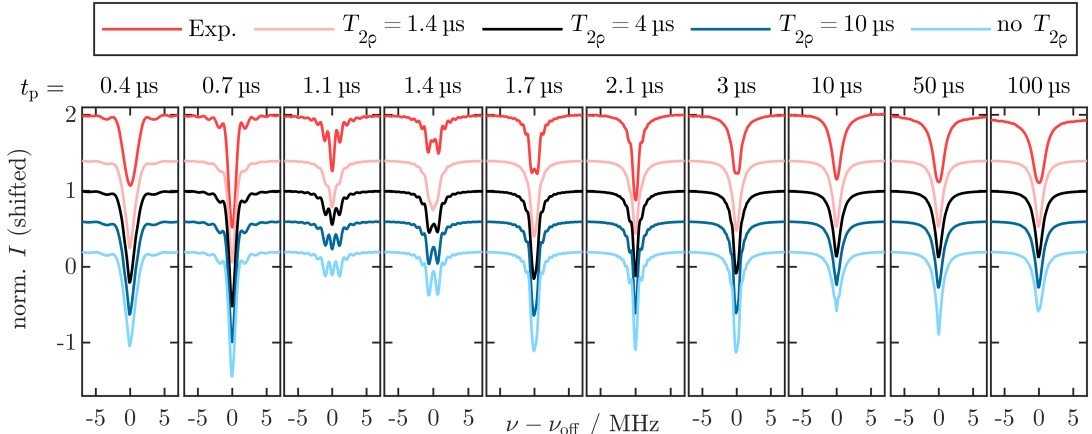

**Figure A15.** Experimental CHEESY $z'$-profiles (red) compared to results from Spinach simulations with different values from $T_{2\rho}$. The spectra were shifted vertically for the sake of clarity.

with the known hyperfine coupling tensor of BDPA ($A$-tensor taken from Weis et al. (2000), see Fig. A16 (b)) led to a distinct splitting (dark red curve), whereas multiple protons resulted in a more homogeneous line shape. Again, this irregular EPR

line shape slightly distorted the simulated CHEESY profiles. Apart from these differences, the CHEESY profiles calculated with a different number of coupled protons are identical (Fig. A16 (c)). This suggests that nuclear coupling does not affect the inversion profiles and can be neglected in the following considerations.

For these expanded spin systems, the relaxation superoperator $\hat{\hat{R}}_{2\rho}$ had to be adapted. Comparison of the performance of different relaxation matrices (data not shown) suggested that using a standard relaxation matrix for transversal relaxation

where the diagonal elements $R_{ij,ij}$ are $T_{2\rho}^{-1}$ if $|i\rangle \rightarrow |j\rangle$ is an electron spin transition is not sufficient to describe all relaxation pathways. Instead, $R_{ij,ij} = T_{2\rho}^{-1}$ was set for all elements $i \neq j$ with all other elements of $\hat{\hat{R}}_{2\rho}$ being zero. This is equivalent to defining an equal decoherence rate for all types of SL coherence. We assume that this becomes necessary because states in the tilted frame are mixed and the nuclear spin partially decoupled. A more thorough investigation of this effect is beyond the scope of this work and requires more analysis in the future.





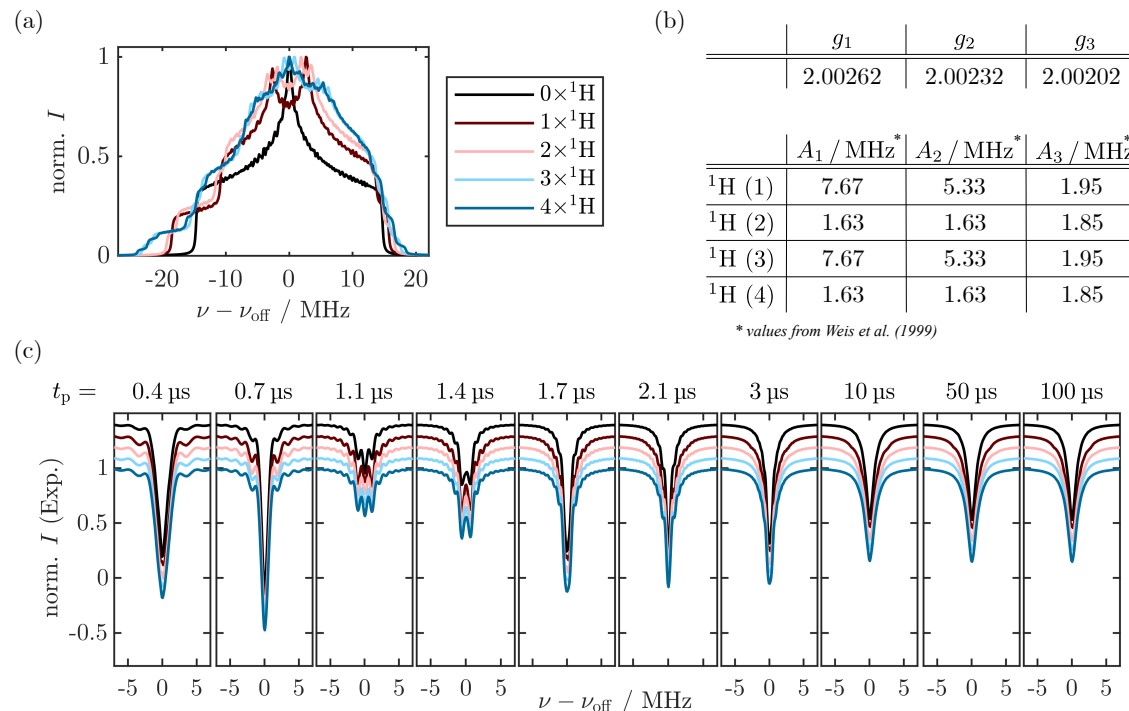

**Figure A16.** (a) Normalized EPR spectra of an electron spin with $g$-anisotropy coupled to up to four proton nuclei simulated using the Spinach library (Hogben et al., 2011). (b) Parameters used for the simulations. Elements of the $g$-tensor were set to arbitrary values to achieve sufficient spectral width. Hyperfine Couplings are taken from Weis et al. (1999) for $^2$H-BDPA and corrected by the ratio of the gyromagnetic ratios of $^1$H and $^2$H. Protons one to four were added successively to obtain the spectra with zero to four coupled nuclei. (c) CHEESY profiles for the spin systems with zero to four nuclei (same color code as in (a)) simulated with the Spinach library. For clarity, the spectra are vertically shifted.

## A14  Frame transformation of the Bloch equations

In order to perform calculations with the Bloch equations in the vector analogue of the tilted frame, we start with the Bloch equations in the rotating frame without relaxation terms (Bloch, 1946):

$$\dot{\boldsymbol{M}}' = \begin{pmatrix} 0 & -\Omega_S & \omega_1 \\ \Omega_S & 0 & 0 \\ -\omega_1 & 0 & 0 \end{pmatrix} \boldsymbol{M}' \tag{A8}$$



where $\boldsymbol{M'} = (M_{x'}, M_{y'}, M_{z'})$ is the magnetization vector and $\dot{\boldsymbol{M}}'$ its time derivative. We apply a rotation around $\boldsymbol{x'}$, i.e.,

perpendicular to the MW field along $y'$, by $\theta = \mathrm{Tan}^{-1}(\omega_1/\Omega_S)$. In the following, we will use the shorthand notations:

$$R_x(\theta) = \begin{pmatrix} 1 & 0 & 0 \\ 0 & \cos(\theta) & -\sin(\theta) \\ 0 & \sin(\theta) & \cos(\theta) \end{pmatrix} \equiv R \tag{A9}$$

$$R_x(-\theta) = R_x^{-1}(\theta) = \begin{pmatrix} 1 & 0 & 0 \\ 0 & \cos(\theta) & \sin(\theta) \\ 0 & -\sin(\theta) & \cos(\theta) \end{pmatrix} \equiv R^{-1} \tag{A10}$$

Multiplication from the left with $R$ and insertion of $R^{-1}R = \mathbb{1}$ yields the evolution of the magnetization vector in the tilted frame $\boldsymbol{M''} = R\boldsymbol{M'}$.

$$\dot{\boldsymbol{M}}'' = R \begin{pmatrix} 0 & -\Omega_S & \omega_1 \\ \Omega_S & 0 & 0 \\ -\omega_1 & 0 & 0 \end{pmatrix} \boldsymbol{M'} = R \begin{pmatrix} 0 & -\Omega_S & \omega_1 \\ \Omega_S & 0 & 0 \\ -\omega_1 & 0 & 0 \end{pmatrix} R^{-1} R \boldsymbol{M'} = R \begin{pmatrix} 0 & -\Omega_S & \omega_1 \\ \Omega_S & 0 & 0 \\ -\omega_1 & 0 & 0 \end{pmatrix} R^{-1} \boldsymbol{M''}$$

$$\tag{A11}$$

To obtain the new evolution matrix, we calculate the matrix product:

$$
\begin{aligned}
R \begin{pmatrix} 0 & -\Omega_S & \omega_1 \\ \Omega_S & 0 & 0 \\ -\omega_1 & 0 & 0 \end{pmatrix} R^{-1} &= \begin{pmatrix} 1 & 0 & 0 \\ 0 & \cos(\theta) & -\sin(\theta) \\ 0 & \sin(\theta) & \cos(\theta) \end{pmatrix} \begin{pmatrix} 0 & -\Omega_S & \omega_1 \\ \Omega_S & 0 & 0 \\ -\omega_1 & 0 & 0 \end{pmatrix} \begin{pmatrix} 1 & 0 & 0 \\ 0 & \cos(\theta) & \sin(\theta) \\ 0 & -\sin(\theta) & \cos(\theta) \end{pmatrix} \\
&= \begin{pmatrix} 1 & 0 & 0 \\ 0 & \cos(\theta) & -\sin(\theta) \\ 0 & \sin(\theta) & \cos(\theta) \end{pmatrix} \begin{pmatrix} 0 & -\Omega_S\cos(\theta) - \omega_1\sin(\theta) & -\Omega_S\sin(\theta) + \omega_1\cos(\theta) \\ \Omega_S & 0 & 0 \\ -\omega_1 & 0 & 0 \end{pmatrix} \\
&= \begin{pmatrix} 0 & -\Omega_S\cos(\theta) - \omega_1\sin(\theta) & -\Omega_S\sin(\theta) + \omega_1\cos(\theta) \\ \Omega_S\cos(\theta) + \omega_1\sin(\theta) & 0 & 0 \\ \Omega_S\sin(\theta) - \omega_1\cos(\theta) & 0 & 0 \end{pmatrix} \\
&= \begin{pmatrix} 0 & -\omega_{\mathrm{eff}} & 0 \\ \omega_{\mathrm{eff}} & 0 & 0 \\ 0 & 0 & 0 \end{pmatrix}
\end{aligned}
\tag{A12}
$$

where the last transformation step is performed with $\Omega_S = \omega_{\mathrm{eff}}\cos(\theta)$ and $\omega_1 = \omega_{\mathrm{eff}}\sin(\theta)$. Insertion into eq. A11 yields the differential equations for the tilted frame magnetization. In this tilted frame expression, we add the transversal relaxation rates



on the matrix diagonals, thus defining the decay of $M_{x''}$ and $M_{y''}$.

$$\dot{\boldsymbol{M}}'' = \begin{pmatrix} -1/T_{2\rho} & -\omega_{\text{eff}} & 0 \\ \omega_{\text{eff}} & -1/T_{2\rho} & 0 \\ 0 & 0 & 0 \end{pmatrix} \boldsymbol{M}'' = \begin{pmatrix} -1/T_{2\rho} \cdot M_{x''} - \omega_{\text{eff}} \cdot M_{y''} \\ \omega_{\text{eff}} \cdot M_{x''} - 1/T_{2\rho} \cdot M_{y''} \\ 0 \end{pmatrix} \tag{A13}$$

This set of differential equations is solved by the standard approach of defining $M_{+''} = M_{x''} + iM_{y''}$.

$$\dot{M}_{+''} = \dot{M}_{x''} + i\dot{M}_{y''} = \omega_{\text{eff}}\left[-M_{y''} + iM_{x''}\right] - 1/T_{2\rho} \cdot \left[M_{x''} + iM_{y''}\right] = \left[i\omega_{\text{eff}} - 1/T_{2\rho}\right]M_{+''} \tag{A14}$$

The solution of this differential equation is an exponential of the form:

$$M_{+''}(t) = \left[M_{x''}(0) + iM_{y''}(0)\right] \cdot \left[\cos(\omega_{\text{eff}}t) + i\sin(\omega_{\text{eff}}t)\right] \cdot e^{-t/T_{2\rho}} \tag{A15}$$

After factorization and separating the expression for $M_{+''}(t)$ into its real and imaginary part, an expression for $\boldsymbol{M}''(t)$ is obtained.

$$\boldsymbol{M}''(t) = \begin{pmatrix} \left[M_{x''}(0)\cos(\omega_{\text{eff}}t) - M_{y''}(0)\sin(\omega_{\text{eff}}t)\right] \cdot e^{-t/T_{2\rho}} \\ \left[M_{y''}(0)\cos(\omega_{\text{eff}}t) + M_{x''}(0)\sin(\omega_{\text{eff}}t)\right] \cdot e^{-t/T_{2\rho}} \\ M_{z''}(0) \end{pmatrix} \tag{A16}$$

The initial conditions in the tilted frame are calculated from the rotating frame magnetization at $t = 0$ which is $\boldsymbol{M}(0)' = (0, 0, M_0)$ so that, after back-transformation into the rotating frame, we end up with an analytical expression for $\boldsymbol{M}'(t)$.

$$\boldsymbol{M}''(0) = R \cdot \boldsymbol{M}'(0) = \begin{pmatrix} 1 & 0 & 0 \\ 0 & \cos(\theta) & -\sin(\theta) \\ 0 & \sin(\theta) & \cos(\theta) \end{pmatrix} \begin{pmatrix} 0 \\ 0 \\ M_0 \end{pmatrix} = \begin{pmatrix} 0 \\ -\sin(\theta)M_0 \\ \cos(\theta)M_0 \end{pmatrix} \tag{A17}$$

$$\boldsymbol{M}''(t) = \begin{pmatrix} \sin(\omega_{\text{eff}}t)\sin(\theta)M_0 \cdot e^{-t/T_{2\rho}} \\ -\cos(\omega_{\text{eff}}t)\sin(\theta)M_0 \cdot e^{-t/T_{2\rho}} \\ \cos(\theta)M_0 \end{pmatrix} \tag{A18}$$

$$\boldsymbol{M}'(t) = R^{-1}\boldsymbol{M}''(t) = \begin{pmatrix} 1 & 0 & 0 \\ 0 & \cos(\theta) & \sin(\theta) \\ 0 & -\sin(\theta) & \cos(\theta) \end{pmatrix} \begin{pmatrix} \sin(\omega_{\text{eff}}t)\sin(\theta)M_0 \cdot e^{-t/T_{2\rho}} \\ -\cos(\omega_{\text{eff}}t)\sin(\theta)M_0 \cdot e^{-t/T_{2\rho}} \\ \cos(\theta)M_0 \end{pmatrix}$$

$$= \begin{pmatrix} \sin(\omega_{\text{eff}}t)\sin(\theta)M_0 \cdot e^{-t/T_{2\rho}} \\ \left[-\cos(\omega_{\text{eff}}t)\sin(\theta)\cos(\theta) \cdot e^{-t/T_{2\rho}} + \sin(\theta)\cos(\theta)\right]M_0 \\ \left[\cos(\omega_{\text{eff}}t)\sin^2(\theta) \cdot e^{-t/T_{2\rho}} + \cos^2(\theta)\right]M_0 \end{pmatrix} \tag{A19}$$





In the limit of long evolution times, all terms with $e^{-t/T_{2\rho}}$ decay to zero and a reduced magnetization vector remains:

$$
\boldsymbol{M}(t \to \infty) = \lim_{t \to \infty}
\begin{pmatrix}
\sin(\omega_{\text{eff}}t)\sin(\theta)M_0 \cdot e^{-t/T_{2\rho}} \\
\left[-\cos(\omega_{\text{eff}}t)\sin(\theta)\cos(\theta) \cdot e^{-t/T_{2\rho}} + \sin(\theta)\cos(\theta)\right]M_0 \\
\left[+\cos(\omega_{\text{eff}}t)\sin^2(\theta) \cdot e^{-t/T_{2\rho}} + \cos^2(\theta)\right]M_0
\end{pmatrix}
$$
$$
=
\begin{pmatrix}
0 \\
\sin(\theta)\cos(\theta)M_0 \\
\cos^2(\theta)M_0
\end{pmatrix}
=
\begin{pmatrix}
0 \\
\frac{(\nu - \nu_{\text{off}})\nu_1}{\nu_1^2 + (\nu - \nu_{\text{off}})^2} \\
1 - \frac{\nu_1^2}{\nu_1^2 + (\nu - \nu_{\text{off}})^2}
\end{pmatrix}
\tag{A20}
$$

where the last step of the rearrangement was done using *MATHEMATICA*.

### A15 Solution of Bloch equations in the rotating frame

When adding the transversal relaxation terms in the rotating frame, Eq. A21 is obtained (Bloch, 1946):

$$
\dot{\boldsymbol{M}}' =
\begin{pmatrix}
\dot{M}_{x'} \\
\dot{M}_{y'} \\
\dot{M}_{z'}
\end{pmatrix}
=
\begin{pmatrix}
-1/T_2 & -\Omega_S & \omega_1 \\
\Omega_S & -1/T_2 & 0 \\
-\omega_1 & 0 & 0
\end{pmatrix}
\begin{pmatrix}
M_{x'} \\
M_{y'} \\
M_{z'}
\end{pmatrix}
\tag{A21}
$$

As the analytical solution of Eq. A21 is rather cumbersome and provides little intuitive insight into the physical processes (Torrey, 1949), we used a numerical differential equation solver implemented in *MATLAB* (*ode45*) to obtain the results displayed in Fig. 7.

### Appendix B: Additional information on the figures in the main text

In the following, the experimental conditions and data processing steps of the figures in the main text are given. This includes a reference to the experimental data sets and processing scripts that are uploaded in a separate data repository (https://doi.org /10.25625/B11CUC). The experimental data sets are provided as *.DTA* and *.DSC* files and were processed with the *eprload* command implemented in *EasySpin* (Stoll and Schweiger, 2006). The processing and simulation files are provided as *MATLAB* code. Additionally, the analysis files for the figures in the supporting information can be found in the data repository.

### B1 Fig. 1 (SL Rabi nutations)

SL Rabi nutations obtained with the sequence in Fig. 1 (a) and the following parameters: $T = 100\,\text{K}$, $t_{\pi/2} = 18\,\text{ns}$, $t_\pi = 36\,\text{ns}$, $\tau = 1\,\mu\text{s}$, $\tau_0 = 1\,\mu\text{s}$, $\tau_1 = 2\,\mu\text{s}$, $a_{\text{PM}} = 0.4$, shot repetition time (SRT) = $40\,\text{ms}$, shots per point (SPP) = 4, scans = 3. A 4-step phase cycle of the $\pi$ refocusing pulse was used:

$$\phi_\pi = [0°, 90°, 180°, 270°]; \quad \phi_{\text{det}} = [+, -, +, -]$$

Rabi traces were obtained by recording the integrated echo intensity as a function of $t_{\text{PM}}$. For the Fourier transformation, the traces were phase- and baseline-corrected and apodized with a half cosine bell function. This apodized data set was zero filled





with twice the number of data points and Fourier transformed using the *fft* command in *MATLAB*. The simulation procedure is illustrated in Fig. 1 (c). The same overall trace length and FT procedure as in the experiment was used for the simulations. *Experimental data sets:* 999a−999f. *Processing file:* SL_Rabi_nutations_analysis.m.

**B2  Fig. 2 (SL echoes and WOP SL echoes)**

SL echoes and WOP SL echoes obtained with the sequences in Fig. 2 (a) and (b), respectively.The following parameters were used: $T = 100\,\mathrm{K}$, $t_{\pi/2} = 18\,\mathrm{ns}$, $t_\pi = 36\,\mathrm{ns}$, $\tau = 0.3\,\mu\mathrm{s}$, $\tau_0 = 2\,\mu\mathrm{s}$, $\tau_1 = 3\,\mu\mathrm{s}$, $\tau_3 = 10.7\,\mu\mathrm{s}-\tau_2$, $a_{\mathrm{PM}} = 0.4$, $\nu_{\mathrm{PM}} = 15\,\mathrm{MHz}$, $t_{\mathrm{PM1}} = t_{\mathrm{PM3}} = 80\,\mathrm{ns}$, $t_{\mathrm{PM2}} = 160\,\mathrm{ns}$, SRT = 40 ms, SPP = $1-4$, scans = 5. In addition to the PCs denoted in the figure, a 2-step PC of PM3 was performed in all cases:

$\phi_{\mathrm{PM3}} = [0°, 180°]; \quad \phi_{\mathrm{det}} = [+, -]$

Echoes were obtained by recording the integrated echo intensity as a function of $\tau_2$. The echo traces were phase corrected so that the dominant signal in the trace without PC of PM2 was maximized in the real part of the signal. The experiments with different PC were subsequently corrected with the same phase factor. *Experimental data sets:* 967a−967f. *Processing files:* SL_echo_analysis.m and WOP_SL_echo_analysis.m.

**B3  Fig. 3 (SL relaxation measurements)**

SL echo decay traces were obtained using the sequence in Fig. 2 (a). The following parameters were used for the trace in Fig. 3 (a): $T = 100\,\mathrm{K}$, $t_{\pi/2} = 20\,\mathrm{ns}$, $t_\pi = 40\,\mathrm{ns}$, $\tau = 0.3\,\mu\mathrm{s}$, $\tau_0 = 100\,\mu\mathrm{s}$, $\tau_1 = \tau_2$, $\tau_3 = 32.2\,\mu\mathrm{s} - \tau_1 - \tau_2$, $a_{\mathrm{PM}} = 0.4$, $\nu_{\mathrm{PM}} = 15\,\mathrm{MHz}$, $t_{\mathrm{PM1}} = t_{\mathrm{PM3}} = 80\,\mathrm{ns}$, $t_{\mathrm{PM2}} = 160\,\mathrm{ns}$, SRT = 40 ms, SPP = 4, scans = 3. The trace was obtained by recording the integrated echo intensity as a function of $\tau_1 = \tau_2$. A 2-step PC of PM3 was performed. The bare state echo decay was obtained

using the Hahn echo sequence ($\pi/2 - \tau - \pi - \tau - echo$) with $t_{\pi/2} = 14\,\mathrm{ns}$, $t_\pi = 28\,\mathrm{ns}$. The experiments in (b) were performed with similar parameters but $t_{\mathrm{PM}}$ was, for each experiment, adjusted to the different SL Rabi frequencies. *Experimental data sets:* 955, 1018b. *Processing files:* T2rho_analysis.m.

**B4  Fig. 4 and 5 (CHEESY $z'$-profiles)**

CHEESY $z'$-profiles were measured using the sequence in Fig. 4 (a) and the following parameters: $T = 100\,\mathrm{K}$, $t_{\pi/2,\mathrm{chirp}} = 1\,\mu\mathrm{s}$,

$t_{\pi,\mathrm{chirp}} = 0.5\,\mu\mathrm{s}$, $\tau = 3\,\mu\mathrm{s}$, SRT = 40 ms, SPP = 150, scans = 1. An 8-step PC of the chirp echo was used:

$\phi_{\pi/2,\mathrm{chirp}} = [0°]_4, [180°]_4; \quad \phi_{\pi,\mathrm{chirp}} = [0°, 90°, 180°, 270°]_2; \quad \phi_{\mathrm{det}} = [+, -]_2, [-, +]_2$

The chirp pulses had a sweep width of 200 MHz centered at the maximum of the resonator dip. The frequency sweep of the chirp pulses was adjusted to compensate for the resonator profile (FWHM $\approx 140\,\mathrm{MHz}$), yielding an approximately constant adiabaticity (Doll and Jeschke, 2014). A wideband, uniform rate, smooth truncation (WURST) amplitude modulation (Kupce

and Freeman, 1995) was applied to yield the final chirp pulse.



Before the experiment, the magnetic field was adjusted so that the BDPA EPR line was centered at $\nu_{\text{off}} = -31\,\text{MHz}$ and the analyzed pulse was applied with the same frequency offset. FT was performed using the *fft* command in *MATLAB* after choosing a FT window of $5\,\mu\text{s}$ symmetrically around the echo, apodization with a cosine bell function and zero filling with a number of zeros equal to the length of the data set. Zero-order phase correction was performed so that the integral of the FT

was maximized in the real part of the signal. *Experimental data sets:* 1025. *Processing files:* CHEESY_z_profile_analysis.m and CHEESY_z_profile_SimSpinach.m.

### B5   Fig. 6 (CHEESY $x'y'$-profiles)

The CHEESY $x'y'$-profiles in Fig. 6 (c) and (d) were obtained using the pulse sequence in (a) and $T = 100\,\text{K}$. The chirp pulses were identical to the $\pi$-chirp pulse in the previous section. The delays were set to $\tau_1 = 1\,\mu\text{s}$ and $\tau_2 = 4\,\mu\text{s}$, together with SRT =

$40\,\text{ms}$, SPP = 250, scans = 1 and an offset $\nu_{\text{off}} = -29\,\text{MHz}$. A 32-step PC was used:

$$\phi_{\text{analyzed}} = [0°]_{16}, [180°]_{16}; \qquad \phi_{\pi,\text{chirp}(1)} = \{[0°]_4, 90°]_4, [180°]_4, [270°]_4\}_2;$$

$$\phi_{\pi,\text{chirp}(2)} = [0°, 90°, 180°, 270°]_8: \qquad \phi_{\text{det}} = \{[+, -]_2, [-, +]_2\}_2, \{[-, +]_2[+, -]_2\}_2)$$

After manual selection of an FT window with $4\,\mu\text{s}$ length, FT was performed as in the previous section. Zero-order phase correction was performed so that the dispersive signal was maximized in the imaginary channel.

The measurement of $T_{1\rho}$ (e) was performed with the sequence in (b), where $t_{\pi/2} = 14\,\text{ns}$, $t_\pi = 28\,\text{ns}$, $\tau = 1\,\mu\text{s}$ and $\nu_{1,\text{SL}} = 0.8\,\text{MHz}$. The intensity was scaled to overlap with the data points from the R-CHEESY experiment. *Experimental data sets:* 875, 876, 968. *Processing files:* CHEESY_xy_profile_analysis.m.

### B6   Fig. 7 (Bloch equations)

The experimental data shown in (a) and (b) is the same as in Fig. 4 (c) and Fig. 6 (c).

*Processing files:* CHEESY_z_profile_analysis.m, CHEESY_xy_profile_analysis.m. and Bloch_simulations.m

*Author contributions.* ML, FH, and MB designed the research. Experiments were set up by ML and NW. ML carried out experiments, data analysis and simulations with help from FH and NW. ML and FH wrote the manuscript, which was edited by NW and MB.

*Competing interests.* The authors declare that they have no conflict of interest.

*Acknowledgements.* We thank Dr. Igor Tkach for his help in the experimental setup of SL experiments at high frequency. We also thank Prof.
Ilya Kuprov for his valuable help in setting up Spinach simulations as well as Prof. Steffen Glaser, Dr. Yvo Pokern, Dr. Roberto Rizzato and Dr. Deniz Sezer for helpful discussions. ML, FH, and MB acknowledge funding from the Max Planck Society and the European Research



Council Advanced Grant *BIO-enMR*. NW was supported by the Swiss National Science Foundation (Postdoc.Mobility grant 206623). FH acknowledges funding from the DFG Walter Benjamin Program (Project HE 9563/1-1).





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
