# Peer review of "Electron spin dynamics during microwave pulses studied by 94 GHz chirp and phase-modulated EPR experiments"

_Magnetic Resonance, 2024_

## Author Response (AR1)

Blue: Answer to comment *(italic: changed paragraph)*
Green: Task completed
*All references to specific lines of the manuscript (L. …) are given* **with reference to the marked-up version of the manuscript**

General comments:

We want to express our gratitude to both of the reviewers for their thorough effort and the helpful and constructive feedback and to the editor for handling the submission process. Before the response to the individual reviewer comments, we will briefly describe the minor additional changes that were made during the revision.

- Prompted by the suggestion of Anonymous Referee #2 to compare our $T_{2rho}$ values to the Carr-Purcell sequence and Hahn echo decays with selective pulses we reevaluated our relaxation data. In particular, we included the dead-time of the Hahn echo sequence (i.e., the minimal $2*tau$). This led to a slight change of the resulting $T_m$ value (see Sect. 4.1.3). Furthermore, we remeasured the Hahn echo decay with non-selective pulses, which led to a minor change of the mono-exponential $T_m$.

- In the original version of the manuscript, there was a typo in the mono-exponential $T_{2rho}$ for $nu_1 = 15$ MHz (3.9 us instead of 3.6 us). The value was corrected (Sect. 4.1.3, L. 396)

- The change in values of $T_{2rho}$ and $T_m$ led to a different ratio of the two. We adapted this ratio in L. 7 of the abstract and in L. 395 of Sect. 4.1.3 by changing it to $2 - 3$. For the comparison of $T_{2rho}$ and $T_m$ for different $nu_1$ we changed the ration to $2 - 4$ (L. 404).

- In L. 119, the expression for $rho''(t)$ was given with time-dependent $U_2(t)$ operators. As $U_2$ is not time dependent, we changed the expression using $U_2$ instead of $U_2(t)$

- In L. 139, a factor ½ was missing in the definition of $rho'(0)$. We changed the expression accordingly.

- In L. 588 (Eq. A1), the expressions for $I(t)$ and $Q(t)$ were mixed up.

- In the caption of Fig. A9, the assignment of the dotted vertical lines was mixed between (a) and (b). We changed the description accordingly

Anonymous Referee #1:

The paper by Lenjer et al. investigates phase-modulated pulses during spin-locking as well as chirp pulses for hole-burning at W-band for a model system consisting of a BDPA radical, including a first demonstration of phase-modulated pulses at this frequency. Their experimental work is complemented by an in-depth analysis of observed effects at several levels of theory, from Bloch equations to advanced spin dynamics simulations. The experimental and theoretical work is carefully performed, the manuscript is well-written, and

the results are presented in detailed and informative figures. The authors helpfully include thorough descriptions of the experimental implementation, and the scripts used to perform the described simulations.

This work covers many interesting aspects and constitutes a valuable addition to the literature. However, the current manuscript is quite long and very dense, which is not too surprising given the complexity of the topic, but, in my view, a couple of small changes could be made to improve readability for a wider audience.

I would recommend considering the following points:

We thank Anonymous Referee #1 for the helpful and extensive comments, as well as for the positive feedback.

- The paper discusses both phase-modulated pulses during spin-locking as well as frequency- and amplitude-modulated chirp pulses, and shows that the theoretical treatment developed for the first case is valuable for the analysis of the excitation profiles of long chirp microwave pulses. The motivation for performing both classes of experiments and the exact connection between them could be made much clearer. In particular in the results, the transition from one to the other is quite abrupt at the moment.

  Thank you for this comment. In fact, we did not measure the excitation profiles of chirp pulses, but used chirp pulses to measure excitation profiles for long, rectangular MW pulses. We specified the motivation for the different types of experiments in the introduction figure (see below) and added the following paragraph to L. 53 of the introduction:

  *While the first class of experiments allows observation of the dynamics during MW irradiation, the second enables observation of the spin state immediately after a MW pulse. Therefore, the two approaches provide complementary information that we could use as a basis for density matrix simulations of an isolated electron spin under MW irradiation.*

  We added an introductory sentence to both section 4.1 (L. 315):
  *We performed SL EPR experiments to probe the dynamics during MW irradiation*
  and section 4.2 (L. 428):
  *CHEESY experiments were performed to analyze the spin state immediately after long, rectangular MW pulses.*

  To improve the transition between SL and CHEESY experiments, we added the following sentence in L. 422:
  *Overall, these experiments provided us with an estimate for the decoherence time during SL over a broad range of MW powers. In the following sections, this will be used to model the spin dynamics during MW irradiation by calculating trajectories of our S=1/2 model system. For this, a transverse relaxation time of T_2rho = 4 us will be assumed as an approximation of the mono-exponential T_2rho time. Due to the strong correlation of T_2rho and xi and the limitation of our simulation method to mono-exponential decays, we refrain from using the stretched exponential results for the following calculations.*

- An overview figure showing all of the pulse sequences employed (before they are discussed in the Materials and Methods and Results sections) and a clearer summary explaining the motivation for each in the introduction would be extremely helpful for making this paper more accessible to a general audience.

  We included the following figure in the introduction:

[Figure]

(a) Spin lock experiments:
Dynamics during MW irradiaton

**SL Rabi nutation:**

→ Pulse optimization

**SL echoes:**

→ Spin dynamics during MW irradiation
with and without preparation

(b) Chirp echo EPR:
Spin state after MW irradiaton

**Chirp echo:**

→ $z'$-profile after long, rectangular
MW pulses

**Refocused chirp echo:**

→ $x'y'$-profile after long, rectangular MW pulses

(c) Simulations

**Home-written
density matrix
simulations:**

initial state $\hat{S}_{z'}$

frame transf.

MW irradiation

$\hat{\rho}''(0) \xrightarrow{\hat{\mathcal{H}}'' t_p} \hat{\rho}''(t_p)$

frame transf.

$\langle \hat{S}_{x',y',z'} \rangle$
final state

Validation

**Spinach simulations:**
→ Electron nuclear
  coupling
→ FT detection

**Bloch equations:**
→ Visualization
* * *
- The thorough explanation of the theoretical basis is very instructive and really makes this section easier to follow for non-experts. For this purpose, I would recommend also explaining more explicitly why the transformation into the tilted frame is necessary for describing spin relaxation in section 2.1.2. It would also be useful to already include the notation of dressed and bare state in the theoretical description section rather than only starting to use it in the Results section.

  Including a clear statement on why calculating relaxation in the eigenframe of the spin Hamiltonian in Section 2 is a useful suggestion. We added a more extensive explanation in L. 152:

  *Note that the effect of R_2rho in Eq.15 is the exponential dampening of the coherent terms rho''_alpha,beta and rho''_beta,alpha in the density vector operator rho''(t) of Eq.12. Only if the density operator is in the eigenbasis of the full spin Hamiltonian, these entries will correspond to pure coherence. Otherwise they correspond to mixtures of population and coherence, which changes the relaxation behavior (vide infra).*

  We added a definition of the bare and dressed state in L. 65:
  *In the following, we refer to a spin system during free evolution as the bare state in contrast to spins subject to continuous MW irradiation, which are in the dressed or SL state.*

- Can the authors specify more clearly what systems their results and insights are applicable for? Is it just for S=1/2 systems with narrow and featureless spectra? In section 3.5, TEMPOL is mentioned as a sample measurements have also been performed on, but it is unclear what the results were and if they matched the results for BDPA (e.g. the surprising low MW intensity for the chirp pi pulse). (At the moment it sounds like the optimization was done on TEMPOL, but in the subsequent paragraph the results seem to be discussed for BDPA?)

Indeed, the chirp nutations were performed using TEMPOL. We decided to do this because, in this way, we could optimize the chirp echo over the whole excitation bandwidth of 200 MHz. Still, nutations with BDPA lead to similar results, including the low nu_1 required for the pi-pulse.

We added a description of the current limitations of FT EPR for TEMPOL in L. 243: *Similarly, the broad EPR spectrum of TEMPOL was perturbed by the FT approach. Within the excitation bandwidth of the chirp pulses, the approximate convolution of the EPR spectrum and the resonator profile was obtained, with some additional distortions caused by pulse imperfections (see Fig. A8).*

The corresponding spectrum can be found in the SI (Fig. A8), together with the following description:

[Figure]

*Fig. A8 shows the analogous comparison for a TEMPOL radical. The spectral width exceeds the sweep range of the chirp pulses by more than 200 MHz, which prevented the detection of the full EPR spectrum by FT spectroscopy. Within the sweep range, the FT spectrum is the approximate convolution of the echo detected EPR and the resonator profile. Additionally, there are some further distortions in the line shape, presumably caused by the pulse edges or reflections in the MW resonator. Although this result shows the limitations of our experimental set-up to completely detect broad EPR spectra, it suggests that FT EPR might be useful for other applications to partially benefit from the multiplex advantage of the FT approach.*

Additionally, we added a paragraph in the Conclusion section (L.555) in which we discuss the transferability of our experimental approach to other samples:
*While we performed the experiments with a BDPA radical which has a narrow and unstructured EPR spectrum, we expect some of the results to be transferable to other samples, including the frequently used nitroxide radicals. First chirp EPR experiments of nitroxide radicals are a promising starting point for using FT EPR for broadband experiments on our W-band spectrometer and the work in this direction is still ongoing.*

- It would be useful to more clearly point out the differences between the simulations performed with home-written routines by the authors and with Spinach, and when and why each is better, i.e. what results can only be reproduced if MW inhomogeneity is taken into account or if powder averaging is included, and how using a Gaussian distribution of resonance offsets compares to modeling the spectrum with artificial g-anisotropy. E.g. it seems like the inclusion of MW inhomogeneity is necessary for the accurate simulation of Rabi nutations, but the corresponding excitation profiles for different pulse lengths seem to no longer require this based on the good agreement of the Spinach simulations with experiment in Fig. 5. Is there a single approach that is

able to model every experimentally observed effect accurately (in SL Rabi nutations, SL echo, Rabi nutations and excitation profiles) and what would that entail?

We thank the referee for pointing out this open question. To clarify the main differences between the home-written and Spinach-based simulations, we changed the description of the Spinach simulation in L. 458 (Sect. 4.2.1):

*We compared our results to simulations using the Spinach library (Hogben et al., 2011) where we explicitly calculated the hole burning in a powder averaged EPR spectrum by Fourier transformation of a time-domain CHEESY signal (see Sect. 3.9 and Sect. A14).*

Indeed, the incorporation of MW inhomogeneity is necessary to reproduce the Rabi decays obtained from the CHEESY z'-profiles quantitatively. We added Figure A15 which shows the experimental Rabi nutation as well as the trace obtained from Spinach simulations, together with an explanation in Sect. 4.2.1 (L. 464):

*However, the dampening of the Rabi oscillation obtained from the intensity at nu – nu_off = 0 was not quantitatively reproduced (compare Fig. A18 and Fig. 5(d)) because increased computational time prevented incorporation of MW inhomogeneity.*

[Figure]

L. 748:

*Simulation of MW inhomogeneity was prevented by the increased computation time of the Spinach simulations. Although the shape of the thus simulated profiles agreed well with the experiments upon visual comparison, the Rabi nutation at nu – nu_off = 0 decayed more slowly in the simulation (see Fig. A18) As described in Sect. A13 the decay of the experimental Rabi nutation is a combination of decoherence and MW inhomogeneity, which cannot be reproduced when omitting the MW inhomogeneity.*

For an evaluation of which simulation method is better suited, we included the following paragraph in L. 474:

*Overall, the Spinach simulations validated the home-written density matrix simulations by demonstrating that, under our conditions, including nuclear couplings has no visible effect on the z'-profiles. At the same time, they highlighted that omitting MW inhomogeneity leads to an underestimation of the Rabi nutation decay. Including the FT step in the simulations improved the agreement between experimental and simulated z'-profiles and allowed a qualitative assessment of the effect of $T_{2rho}$ on reaching dampening of the sinc shape features of the profiles. However, the Spinach simulation requires longer computation time, preventing fast calculation and inclusion of MW inhomogeneity. This is the advantage of the home-written density matrix simulation, which we used as the main computational tool in the following.*

Minor comments and typos:

- Both echos and echoes are used in different parts of the manuscript.
Changed

- In the introduction, where decoherence during spin locking is discussed, it might be useful to include a reference to 10.5194/mr-2024-17 .
We fully agree with this comment. The paper in question was published after first submission of our manuscript, so there was no chance of including it before. We added the reference in L.36 and used it to highlight the difference between a phenomenological treatment of relaxation by rate equations and the more rigorous treatment by Jeschke et al:
*Very recently, Jeschke et al. (2024) calculated electron $T_{2\rho}$ decays not using rate equations but from both the analytical pair product approximation and cluster correlation expansion. Their results show that a rigorous, quantitative treatment of decoherence during spin lock is currently challenging. Therefore, phenomenological treatment of relaxation using rate equations is still highly relevant to describe spin locking experiments.*

- p. 3, line 57, their respective interaction frames
Changed

- p. 3, line 58, clarify this is sufficient for the simple system used in this paper, unless this is really general.
We rephrased the sentence accordingly.

- Consider adding the transformation operator for the transformation into the nutating frame before equation 6 for completeness.
We added the transformation operator in L.91. To keep the name of the transformation operator into the tilted frame (Eq. 7) consistent with the first version of the manuscript and not having to change it into U_3, we named this transformation operator U_PM

- Hyphenation in so-called, spin-locked, Boltzmann-populated.
Changed

- In the paragraph after equation 13, consider starting from the general expression for rho(0) (missing ') and then explaining what has been used in the calculation and why.
We agree that this change will improve the readability of the paragraph. The modified paragraph includes the expression for both rho(0) and rho'(0). Furthermore, we added two sentences that explain which part of the equilibrium density matrix we use for calculations and the transformation of rho'(0) into the tilted frame is shown (L. 141):
*1/2 does not evolve and is neglected in the following. In the tilted frame, the equilibrium density matrix transforms to rho''(0)=U_2 rho'(0) U_2* = cos(theta) S_z'' – sin(theta) S_y''*

- p. 6, line 144, non-resolved -> unresolved
Changed

- In section 3.2, it would be useful to specify how exactly LO leakage is minimized on the spectrometer. Does it include active compensation of the amplitude imbalance of the I and Q channels of the SpinJet AWG?

We rephrased the description accordingly (L; 171):
*As the AWG frequency is centered at zero, active compensation of the input amplitude imbalance and filtering are applied to minimize the contribution of…*

- Inconsistency between use of MW and mw.
Changed

- p. 8, line 213, featured -> structured?
Changed

- p. 8, line 222, these non-ideal conditions
Changed

- p. 9, line 237, Bodenhausen
Changed

- p. 9, line 245, prohibited -> prevented
Changed

- p. 13, line 362, turning angles ... were not optimized ...
Changed

- p. 15, line 382, microwave power dependence
Changed

- Fig. 4, (c) and (d) labels swapped? Wrong part of figure referenced in section A12 of the SI.
Thank you for this comment. We agree that the labeling in the figure is somewhat unintuitive because a,b,c,d does not follow a clear visual pattern. However, we decided to keep the order, as this is how the different panels come up in the text. We changed the reference in Section A.13 (Sect. A12 in the first version of the manuscript) to Fig. 4 (d).

- In the captions to Fig. 4 and 5 it would be useful to mention that experiment and simulation are offset, the slightly offset y scales are not immediately obvious. Also, is this really necessary or could the data also be shown with matching scales?
We added the comments in the figure caption. Without the vertical shift, the profiles overlap significantly, so that visual distinction becomes difficult.

- Legends or clear labels for the different colored lines in the Fig. 4 and 5 would be useful rather than having to rely on the caption (in particular for the light blue lines).
Changed

- Specify the type of fit used to model saturation effects (shown in Fig. A3).
We added the fit type (second order polynomial f(x) = a + bx + cx^2) to the figure caption

- Including an intermediate graph would be useful in Fig. A5, to more clearly and visibly show how the probability distribution is obtained from the recorded experimental data. Also, using nu_PM instead of nu_set might be clearer and more consistent with the rest of the manuscript.
We thank Anonymous Referee # 1 for this helpful comment, which makes the description of the procedure much clearer. We modified the figure accordingly:

(a)

[Figure]

(b)

[Figure]

We deliberately used nu_set and not nu_PM because, while nu_PM is varied in the experiment, nu_set is the position of the maximum of P(nu_1), which is constant within one experiment.

- It would be useful to also include the real part of the signal in Fig. A 10 again in the same figure (in addition to being shown in the main text).
Changed

- Fig. A14 caption: omega -> nu
Changed

- p. 38, line 688: is/was overestimated
Changed

- Fig. A 16, inclusion of the experimental spectrum in (a) would be useful
Changed

Referee 2:

Lenjer et al. present an investigation of electron spin dynamics and relaxation behavior under microwave irradiation at 94 GHz. Periodic phase modulations are used to manipulate locked spins and chirp echo EPR spectroscopy (CHEESY) excitation profiles are measured. All experiments are performed on a commercial spectrometer and the manuscript gives a good impression of the applicability of pulse shaping on this instrument. Accompanying simulations are presented.

The work is suitable for publication in Magnetic Resonance, but it would be appropriate to address the following points.
We thank Anonymous Referee #2 for the helpful suggestions and appreciation of our work.

- The expression for the transformation to the nutating frame should be given.
The expression was added in L. 91 (see answer to AR #1)

- In the experiments without a preparation pulse (Figure 2d), the phases of individual signals do not reproduce if the experiment is repeated. Could the cause be instability of the microwave excitation? Savitsky et al. (https://doi.org/10.1016/j.jmr.2014.02.026) characterized the phase-noise and the frequency-stability of the 84.5 GHz PLL Gunn diode in their Bruker ElexSys E680 spectrometer and found it advantageous for long-term EPR experiments to replace it with a dielectric resonator oscillator. The 84.5 GHz Gunn is likely the source of the phase noise shown in Figure A2.
We are aware that the phase noise of our set-up likely arises from the old reference source, but we thank the referee for pointing out this useful reference and confirming our assumption. We have already purchased a new Gunn diode, but are currently waiting for delivery. However, we do not think this plays a major role in the phase distortions in figure 2d. If this was the case, we would expect problems when averaging multiple scans of SL experiments, which is not the case. We assume the instability is caused by the difference between effective and calibrated turning angle of the PM pulses during signal generation that leads to instabilities in the signal phase.

- The discussion of the echoes observed without a preparation pulse on p. 13 is hard to follow. Is it possible that these echoes are due to off-resonance effects? If omega_eff is near parallel to the z-axis, preparation pulses are typically not needed for a "spin-lock".
We are convinced that the echoes without a preparation pulse are not caused by off-resonance effects but by coherent refocusing of the components perpendicular to the locking field. Indeed, we see the WOP signals in our simulations where the electron spin offset Omega_S is set to zero, excluding all off-resonance effects. Under this assumption, we can assign all signals in Figure 2d to coherence pathways where the initial coherence order is +/- 1 at a time zero.

We agree that there are locked spins where omega_eff is approximately parallel to z. However, these are likely outside of the excitation bandwidth of the PM pulses which, in the off-resonance case, would require a matching of omega_eff >> omega_1 with omega_PM.

- In Figure 3 on p.15 the T_2rho is plotted as a function of nu_1. The authors note that the results agree well with low-field results by Wili et al. Why is this expected? Wili et al. investigated bis-trityl rulers, not BDPA in polystyrene, at considerably lower field and with much higher Rabi frequency.
We thank the referee for pointing out this question. Considering the recent publication by

Jeschke et al. (https://doi.org/10.5194/mr-2024-17) we believe that the good quantitative agreement between the two very different experimental cases is coincidental. While the general effect of nu_1 decoupling the electron spin from its nuclear surroundings is universal, we do not see any reason why this effect should be so similar in the case of the two very different experimental conditions. However, the general trend that T_2rho is longer than T_m is expected to be general. We rephrased the passage accordingly by emphasizing a 'similar trend' in the two experiments (L. 406 in Sect. 4.3.1)

- How do the T_2rho values compare to T_m measured with a CPMG sequence?
We measured the CPMG3 and CPMG4 experiment (N=1 and 2 in the figure below, respectively) and compared to the Hahn echo decay (N=0). As expected, the relaxation time increases with increasing N, but is still smaller than T_2rho.
L. 413 in Sect. 4.1.3:
*Additionally, we measured the echo decay with the 3- and 4-pulse Carr-Purcell (CP) sequences (Carr and Purcell, 1954). Both CP3 and CP4 led to an increase of decoherence time by around 50 % relative to the two-pulse measurement with the same nu_1 (see Sect. A11).*

We added the experimental data in the SI (see below).

[Figure]

L. 705:
*We measured decoherence times using the Carr-Purcell sequence with three and four pulses (CP3 and CP4). The results are shown in Fig. A15 and the decay traces can be found in Fig. A13 (d – f). For both experiments, the decays were well reproduced both by a mono- and a stretched exponential and an increase of decoherence time was observed compared to the two pulse Hahn echo experiment.*

- According to Figure 3b, the value of T_m (measured with a Hahn echo) is independent of the Rabi frequency. Is instantaneous diffusion negligible?
We thank the referee for highlighting this important point. Indeed, further experiments with different pulse lengths of the echo detection pulses showed a significant effect on T_m. The new version of Fig. 3b which includes this data is shown below.

[Figure]

L. 412:
*Analysis of Hahn echo decays measured with lower nu_1 (i.e., longer and more selective pulses) revealed increased T_m times (blue in Fig. 4 (b)), indicating a contribution of instantaneous diffusion to the bare state decoherence.*

This shows that, while the standard T_m value obtained with non-selective pulses is 2 – 4 times smaller than the T_2rho values, selective pulses lead to longer T_m. As the non-selective experiment is usually used to measure T_m, we focused our discussions on this value.

- The reason for the tilting of the frame in the Spinach simulations has to be explained better. Tilted interaction frames are commonly used to derive an effective Hamiltonian. But since Spinach executes the propagation numerically in small time steps, this does not seem to be the reason here. Is the purpose to get the correct relaxation mechanics?
Yes, the frame transformation is necessary to get the correct relaxation dynamics. We stressed that with a more detailed explanation in L. 152 (see comments of AR #1). The purpose is to exclusively dampen the coherence elements of the density matrix by R_2rho. When the spin system is not in the eigenframe, the density matrix elements no longer correspond to eigenstates of the system, leading to mixing of coherent and population terms. Dampening these mixed terms will ultimately lead to zero for all density matrix entries.

To highlight that relaxation is the reason why calculations in the tilted frame are necessary, we repeated the simulations with varied decoherence time in the rotating frame. We modified Fig. A19 accordingly and added an explanation in L. 759 (Sect. A14):

(a) Tilted frame:

[Figure]

(b) Rotating frame (using holeburn.m):

[Figure]

*The analogous simulations with T_m in the rotating frame are shown in Fig. A19 (b). While the profiles without relaxation (light blue) agree quantitatively with the tilted frame results, the profiles with relaxation show broadening for large t_p, which is not observed in the experiment. This deviation increased with decreasing decoherence time. Again, this demonstrates that relaxation is the reason why calculations need to be performed in the tilted frame.*

- In Eq. A2 a subscript p is missing after Delta t. I suggest to remove the exclamation mark as in some programming languages != means not equal to.
Changed